# Differential regulation of mRNA stability modulates transcriptional memory and facilitates environmental adaptation

Bingnan Li [1,2], Patrice Zeis [3,7], Yujie Zhang[2], Alisa Alekseenko[2], Eliska Fürst [4], Yerma Pareja Sanchez[2], Gen Lin [3,8], Manu M. Tekkedil[3], Ilaria Piazza[4], Lars M. Steinmetz[3,5,6] & Vicent Pelechano [2] ✉

Transcriptional memory, by which cells respond faster to repeated stimuli, is key for cellular adaptation and organism survival. Chromatin organization has been shown to play a role in the faster response of primed cells. However, the contribution of post-transcriptional regulation is not yet explored. Here we perform a genome-wide screen to identify novel factors modulating transcriptional memory in *S. cerevisiae* in response to galactose. We find that depletion of the nuclear RNA exosome increases *GAL1* expression in primed cells. Our work shows that gene-specific differences in intrinsic nuclear surveillance factor association can enhance both gene induction and repression in primed cells. Finally, we show that primed cells present altered levels of RNA degradation machinery and that both nuclear and cytoplasmic mRNA decay modulate transcriptional memory. Our results demonstrate that mRNA post-transcriptional regulation, and not only transcription regulation, should be considered when investigating gene expression memory.

Transcriptional memory is the process by which previously expressed genes remain poised for faster reactivation[1,2]. The ability to "remember" previous stimuli and respond faster to future events is key for cellular adaptation and organism survival. For example, in *Saccharomyces cerevisiae* it is well known that previous exposure to different carbon sources[3–5] or stress conditions[6] facilitates future gene expression changes. A well-studied case in budding yeast is the galactose-induced transcriptional memory that leads to faster reactivation of the yeast *GAL* genes[4,7,8]. Transcriptional memory also plays a role in environmental stress adaptation in plants[9,10] and facilitates faster interferon-γ response in humans[11]. In the disease context, transcriptional memory contributes to trained immunity in macrophages, where previous exposure to LPS or ß-glucan can modulate immune tolerance via epigenetic reprogramming[12,13], and also explains how

prior inflammation can modulate tissue regeneration[14]. Thus, the ability of cells to alter future gene expression based on previous stimuli is a fundamental phenomenon in biology and key to understanding cell identity and disease progression.

Multiple mechanisms have been implicated in transcriptional memory, most of which act at the chromatin level (reviewed in[15]). Mechanisms associated with transcriptional memory include chromatin remodeling[7], incorporation of the histone variant H2A.Z or presence of H3K4me2[16–18], association to the nuclear pore, DNA topology[4,16,19] or even long noncoding RNA (lncRNA)-based formation of R-loops[20]. Although we know that many factors are involved in transcriptional memory, our knowledge of this process is far from complete. This is especially true regarding non-chromatin-related factors modulating transcriptional memory. As an example of a factor

[1]Department of Diagnostics, Medical Integration and Practice Center, Cheeloo College of Medicine, Shandong University, Jinan, Shandong 250012, China. [2]SciLifeLab, Department of Microbiology, Tumor and Cell Biology, Karolinska Institutet, Solna, Sweden. [3]European Molecular Biology Laboratory (EMBL), Genome Biology Unit, Heidelberg, Germany. [4]Max Delbrück Center for Molecular Medicine in the Helmholtz Association (MDC Berlin), Berlin, Germany. [5]Stanford Genome Technology Center, Stanford University, Palo Alto, CA, USA. [6]Department of Genetics, School of Medicine, Stanford University, Stanford, CA, USA. [7]Present address: Institute of Molecular and Clinical Ophthalmology Basel, Basel, Switzerland. [8]Present address: AbbVie Pte Ltd, Singapore, Singapore. ✉e-mail: vicente.pelechano.garcia@ki.se

independent of chromatin organization, the cytoplasmic accumulation of *GAL1* (galactokinase 1) protein can facilitate faster response in galactose in primed yeast cells by acting as a positive transcriptional regulator of *GAL* genes[21,22].

To identify novel factors controlling transcriptional memory in *S. cerevisiae*, here we performed a genome-wide screen combining flow cytometry with high-throughput sequencing. We identify multiple genes whose depletion leads to altered gene expression dynamics and focus on the investigation of *RRP6*, a component of the nuclear exosome. The nuclear exosome plays an important role in gene expression and participates in transcription termination[23], modulates transcription directionality[24–26] and even controls the level of enhancer RNA (eRNAs) in mammals[27]. Like other nuclear RNA degradation pathways, in generally it is thought that susceptibility to degradation by the nuclear exosome is set co-transcriptionally[28]. Although traditionally nuclear decay was thought to affect mainly non-coding transcripts[29], recent evidence shows that changes in nuclear RNA degradation rates also facilitates gene expression reprograming in response to stress[30,31]. To dissect the potential contribution of the nuclear exosome to transcriptional memory, we characterize gene expression dynamics in wild-type and *rrp6Δ* strains in response to galactose treatment. We categorize genes based on their ability to promote transcriptional memory for both induction and repression of expression in response to galactose. Next, comparing naive and galactose-primed cells, we investigate the potential contribution of non-coding RNA transcription and chromatin organization to transcriptional memory. We further study the possible direct role of the nuclear exosome modulating mRNA abundance during transcriptional memory. We investigate if the differential binding of nuclear surveillance factors could contribute to the faster gene expression reprograming observed in primed cells. Using RNA-crosslinking data, we show that genes with transcriptional memory display a distinct intrinsic association with nuclear surveillance complexes associated with the nuclear exosome. Next, we investigate the proteome differences between naive and primed cells, focusing on the alteration of the mRNA degradation machinery. Finally, we use RNA metabolic labeling to investigate differences in mRNA turnover between naive and primed cells in multiple mRNA degradation mutants. Our results show that changes in nuclear and cytoplasmic mRNA stability between states contribute to differential gene-expression response and transcriptional memory.

## Results

### Genome-wide screening for factors modulating transcriptional memory

To identify genes able to modulate transcriptional memory, we constructed a reporter system able to track this process. As a model for transcriptional memory, we used the expression of *GAL1* in response to galactose in *S. cerevisiae*. Our memory reporter system contains a fast-folding GFP (sfGFP) under the control of the *GAL1* promoter (*pGAL1*), which has been shown to independently display transcriptional memory[4], and a constitutively expressed mCherry under the control of pTEF1 (see methods and Supplementary Fig. 1A). Since protein accumulation is delayed in respect to mRNA accumulation, and proteins are in general more stable, we modified both fluorescent reporters with degron signals to facilitate the identification of dynamic changes. The addition of the degron signals increased the resolution of our system by decreasing protein stability and thus minimizing the lag time between mRNA and protein accumulation. We first confirmed the ability of this system to measure transcriptional memory in response to galactose (Supplementary Fig. 1B). Both raffinose (non-repressed) and glucose (repressed) media have been used as starting point to investigate transcriptional memory in response to galactose[3,4]. Here we decided to use glucose media (YPD), as this simplifies the genome-wide comparison of gene expression between naive and primed cells (see below).

Next, we transformed the barcoded *S. cerevisiae* deletion collection[32] with our reporter system, isolated cells based on their *pGAL1-sfGFP* expression using flow cytometry and identified each strain using high-throughput sequencing (see methods for details). To control for technical variability, we included 8 wild-type control strains containing strain-specific barcodes. We grew the pooled barcoded collection in rich glucose media (YPD) and transferred exponentially growing cells ($OD_{600}$ 0.4–0.6, $t_0$) to rich galactose media (YPGal) for 3 h ($t_{60}$, $t_{120}$, and $t_{180}$). After this, we transferred cells back to YPD and collected cells after 1.5 ($t_{90GLU}$) and 3 h ($t_{180GLU}$, which served also as time 0 prime, $t_0'$). Finally, we re-exposed cells to galactose for 3 h and collected a sample each hour ($t_{60}'$, $t_{120}'$, and $t_{180}'$). We sorted cells according to their relative sfGFP expression and identified each strain by sequencing their unique barcode. After correcting for the total number of cells, we generated a virtual *GAL1* gene expression profile over time for each strain (Fig. 1A). By comparing the accumulation of sfGFP between naive and primed cells we identified potential modulators of *GAL1* transcriptional memory (Supplementary Data 1, see methods for details).

We identified 35 mutants with decreased transcriptional memory. Those included mutants for the expected chromatin remodeller ISW2[7], as well as components of the THO and TREX complexes (*MFT1* and *THP2*) and mitochondrial factors (*ATP18*, *ATP19*, *FMP30*…) among others (Supplementary Fig. 1C). Interestingly, we also identified 37 mutants in which transcriptional memory is enhanced. These included genes involved in nuclear RNA degradation (*RRP6*, *LRP1*, and *PAP2*) and genes involved in meiosis and cell division (*PMS1*, *MSH6*, *DSE1*…) (Fig. 1B, Supplementary Fig. 1D). Our screen also identified that the depletion of *ELP4* (a member of the elongator complex) enhanced transcriptional memory (Supplementary Fig. 1D), in agreement with what has been recently described[33]. We decided to study the role of RNA degradation depletion in enhancing transcriptional memory, as it was the unique significantly enriched KEGG term (FDR < 0.0432, using default STRING enrichment). We focused on the role of nuclear mRNA surveillance, since two components of the nuclear exosome (*RRP6* and *LRP1*) were identified in our screen (Fig. 1C). Although the nuclear exosome is well-known to control the abundance of Cryptic Unstable Transcripts (CUTs) in yeast[24,25], more recent evidence shows that changes in nuclear degradation rates help to rapidly reprogram gene expression after glucose deprivation[30] and that nuclear decay can dampen the accumulation of full-length mRNAs from stress-responsive genes[31]. As the potential role of nuclear mRNA degradation has not been explored in the context of transcriptional memory, we focused on this new avenue.

### Differential transcriptional memory response in absence of functional nuclear exosome

After identifying a potential role of nuclear RNA decay in transcriptional memory using an ectopic reporter system based on protein expression, we validated this result by measuring mRNA expression from the native *GAL1* locus. First, we determined the optimal sampling points to capture the induction/re-induction kinetics at mRNA level by RT-qPCR (Supplementary Fig. 2A). As expected, optimal RNA sampling points were advanced in comparison to the ones used in our protein-based screen (Supplementary Fig. 2A). As the nuclear exosome has a widespread effect on gene expression[30], we hypothesized that *RRP6* could modulate transcriptional memory also in genes other than *GAL1*. We performed RNA-Seq in both the wildtype (BY4741) and the *rrp6Δ* strain. We measured mRNA abundance during exponential growth in YPD ($t_0$, naive) and 30 min, 1 h and 3 h after change to galactose ($t_{30}$, $t_{60}$ and $t_{180}$). Then we transferred cells to YPD for 3 h ($t_0'$, primed) and measured galactose reinduction at 15 min, 30 min and 1 h ($t_{15}'$, $t_{30}'$ and $t_{60}'$) (Fig. 2A and Supplementary Data 2). naive and primed cell response to galactose was very similar, but primed cells were clearly faster in adapting their transcriptome to the use of galactose as carbon

source (Fig. 2B, Supplementary Fig. 2B) in agreement with the existence of transcriptional memory. Specifically, $t_0$ and $t_0'$ cluster together, while gene expression at $t_{15}'$, $t_{30}'$ and $t_{60}'$ in primed cells clustered with $t_{30}$, $t_{60}$ and $t_{180}$ in naive cells respectively.

Focusing on wild-type cells, we identified 882 genes (ORF-Ts, coding transcriptional units as previously defined[25]) induced in response to galactose in both naive ($t_{180}$) and primed ($t_{60}'$) cells (fold-change > 0, p-adj <0.001) (Supplementary Data 2 and Supplementary Fig. 2C). Induced genes contained genes involved in galactose response but also genes related to oxidative phosphorylation and aerobic respiration (Supplementary Fig. 2D). We also identified 1067 genes which were repressed in response to galactose in both naive and primed cells (fold-change <0, p-adj <0.001) (Supplementary Data 2 and Supplementary Fig. 2C). Repressed genes include genes associated to ribosome biogenesis, translation initiation, and elongation (Supplementary Fig. 2E). Next, we analyzed differential expression along the time course (Supplementary Fig. 2C and Supplementary Data 2). We quantified transcriptional memory by comparing gene expression dynamics in primed versus naive cells. Transcriptional memory is commonly used to refer to genes undergoing transcriptional induction, however, transcriptional memory also impacts gene repression[34]. Thus, we refer to genes as possessing either induction or repression memory (instead of using the general term of transcriptional memory). We classified genes with induction memory as those with higher induction in primed cells compared to naive at either 30 minutes or 1 h

(i.e., induction fold-change in primed state more than 1.5x times the fold-change measured in naive state). We identified 546 genes with induction memory, including *GAL1* as well as other genes related to ATP metabolic process (Supplementary Fig. 2F and Supplementary Data 2). On the other hand, we identified 773 genes with repression memory (i.e., repression fold-change in primed state more than 1.5x times the fold-change measured in the naive state). Repression memory genes were clearly enriched for rRNA processing and ribosome assembly (Supplementary Fig. 2G and Supplementary Data 2). Despite gene expression between naive ($t_0$) and primed ($t_0'$) conditions being similar (Supplementary Fig. 2C), using external spike-ins we determined that primed cells have a higher proportion of mRNA contribution to the total RNA amount, driven mainly by rRNA abundance (Supplementary Fig. 2H). This suggests that after 3 h in glucose, primed cells have adapted their relative mRNA levels to growth in YPD, while rRNA abundance (which has a slower RNA turnover and thus requires more time to reach equilibrium) lags behind.

After investigating mRNA abundance changes in the wild-type strain, we performed the same analysis in *rrp6Δ*. Gene expression changes upon galactose addition in the *rrp6Δ* strain were very similar to the ones observed in a wild-type strain (Supplementary Fig. 2B). We identified 1026 genes (ORF-Ts) induced in response to galactose in both naive (at 180 min) and primed (at 60 min) *rrp6Δ* cells (fold-change > 0, p-adj <0.001), and 1177 genes repressed in response to galactose in both naive and primed cells (fold-change <0, p-adj <0.001)

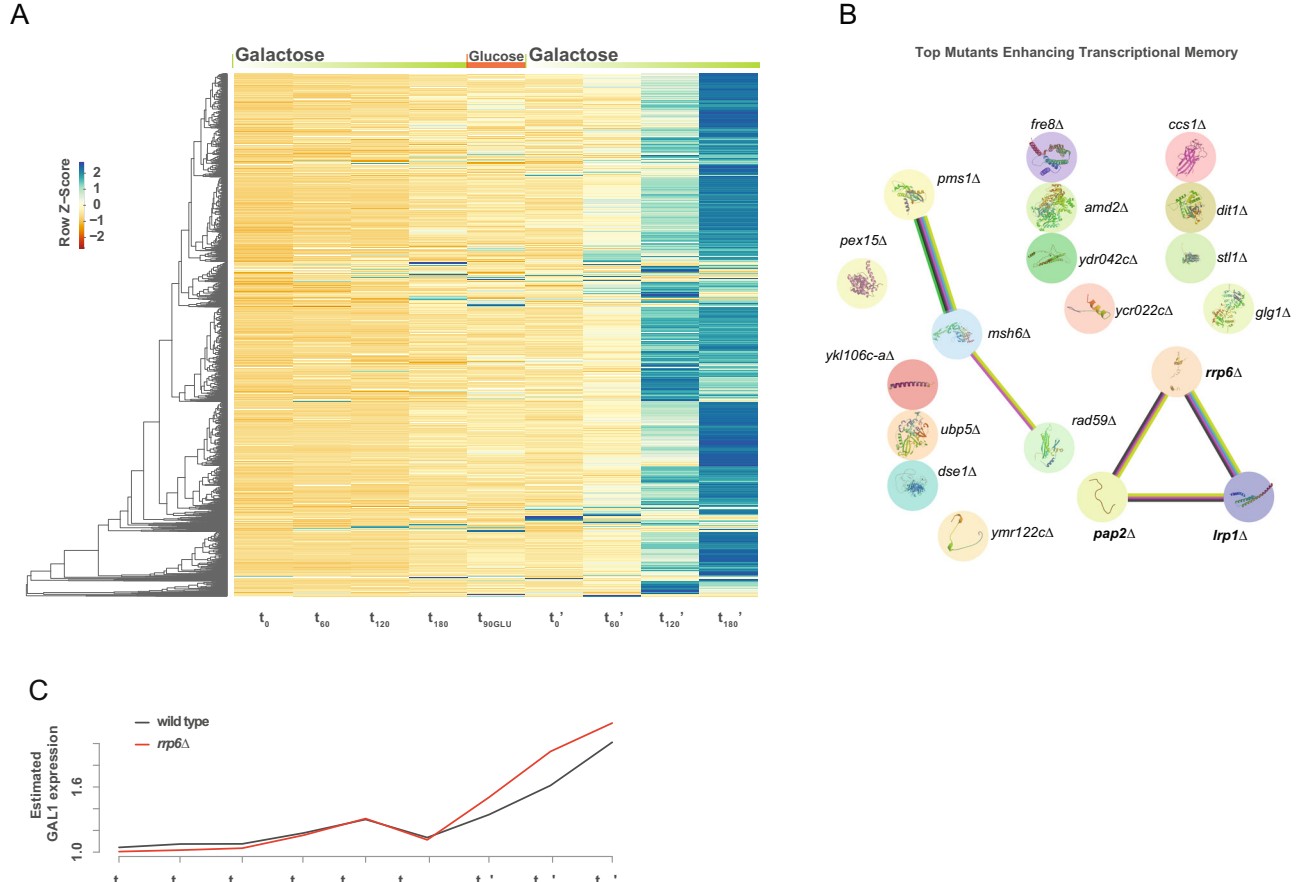

**Fig. 1 | Screen for genetic factors controlling transcriptional memory.**
**A** Heatmap depicting strain-specific z-cores for *pGAL1-sGFP* expression at different time points. Green refers to high sfGFP expression (see Supplementary Data 1 and methods for detail). **B** Top 19 candidate deletion strains displaying enhanced transcriptional memory. Lines represent known interactions as curated by STRING v11.5[48].blue indicate from curated databases, pink indicate experimentally

determined, green indicate predicted interaction as gene neighborhood, light green indicate source as text mining, black indicate co-expression. **C** Relative sfGFP expression for components of the nuclear exosome in red line (*rrp6Δ*) relative to wild type in black line (y-axis from 1 (all cells with no detectable GFP) to 4 (all cells with maximum GFP), see Methods for detail).

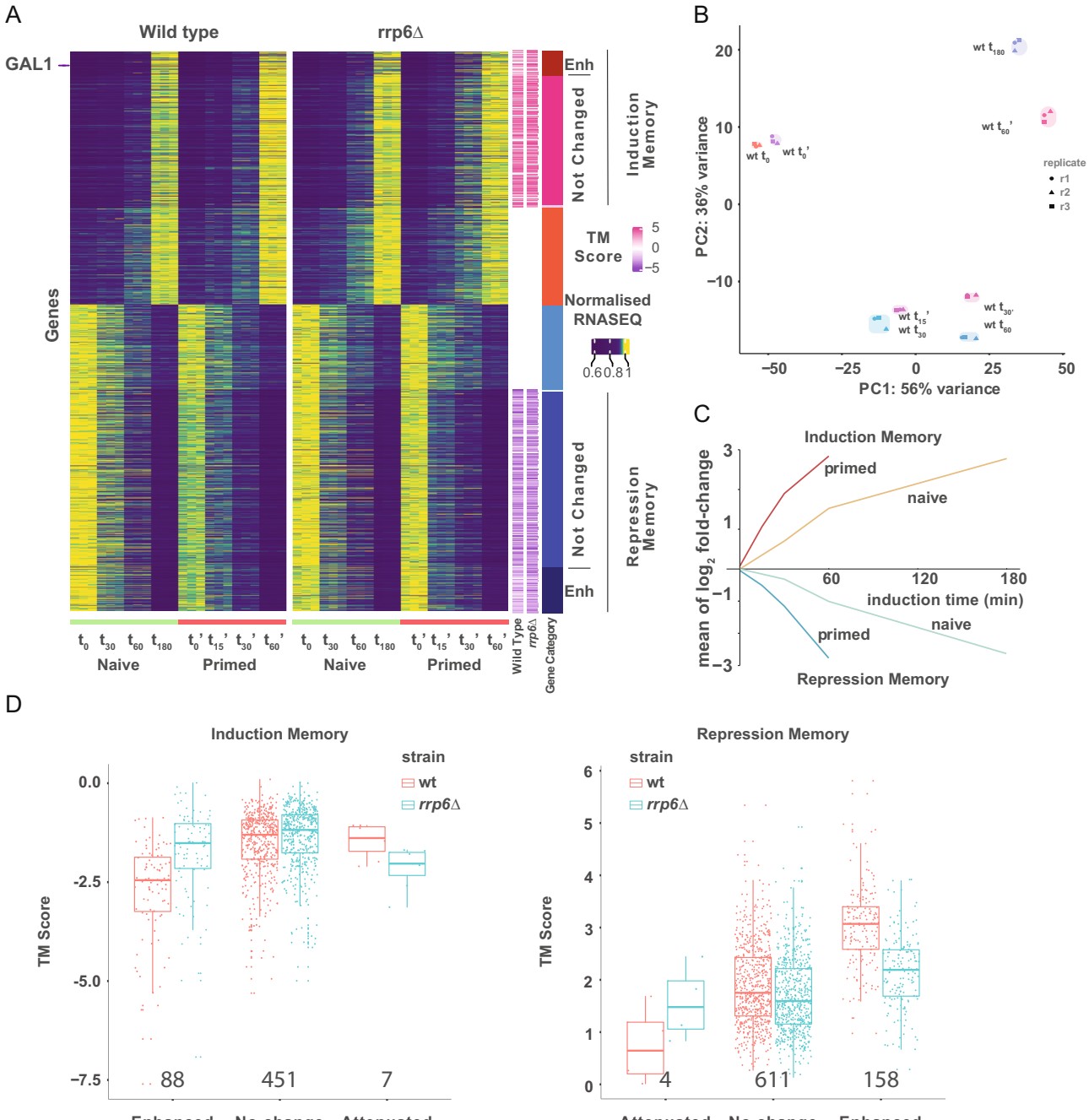

**Fig. 2 | Genome-wide identification of genes with transcriptional memory.**
**A** Heatmap of variable genes in RNA-Seq of both wild type and *rrp6Δ* strains.
Heatmap intensity represents relative mRNA abundance for each replicate at different time points. Gene intensity is normalized by the maximum mRNA abundance across all time points. Gene-specific *transcription memory score* (TM$_{score}$: log$_2$ fold-change t$_{15'}$ vs t$_{180}$) is shown in purple to pink for each strain. The rightmost column shows gene category according to TM$_{score}$: induction memory enhanced in *rrp6Δ* (dark red), induction memory unchanged in *rrp6Δ* (red), induced memory attenuated in *rrp6Δ* (pink, very few genes), induced gene with no memory (orange), repressed gene with no memory (light blue), repression memory attenuated in *rrp6Δ* (turquoise, very few genes), repression memory unchanged in *rrp6Δ* (blue), and repression memory enhanced in *rrp6Δ* (dark blue). 5475 genes examined over 3 independent biological experiments. **B** Principal component analysis (PCA) of mRNA expression (normalized RNA-Seq data) for both first and second induction in wild type. Only coding mRNAs (ORF-Ts) were considered. **C** Mean of log$_2$ fold-

change for genes classified as induction and repression memory genes. Gene expression dynamics in both naive and primed states are shown. **D** Memory Score (TM$_{score}$) for genes classified as induction or repression memory comparing the log$_2$ fold changes lfc$^{15min'}$ – lfc$^{3h}$. Please note that, as we are comparing the early time points (15min') in primed cells to a late time point in naive cells (3h), TM$_{score}$ for genes with induction memory are negative. For repressed genes, as the log$_2$ fold change is negative, a smaller value indicates stronger repression. TM$_{score}$ should be used as a relative measure of gene-specific memory across strains. Box-plot represents the first, second, and third quartile. 546 genes were examined over three independent biological experiments for induction memory. A total of 773 genes were examined over three independent biological experiments for repression memory. The number of analyzed genes are displayed below the box-plots. The first quantile, median, and third quantile are defined as the minimum, center, and maximum bounds of the box-plots.

(Supplementary Data 2A). While the main difference between strains relates to the accumulation of CUTs in *rrp6Δ* (Supplementary Fig. 2I, Supplementary Fig. 2J)[24,25], we also identified 540 genes differentially expressed between wild-type and *rrp6Δ* strains in the naive state (p-adj <0.001, Supplementary Fig. 2K, L). Interestingly, upregulated genes overlapped with the genes classified with induction memory (Hypergeometric test, $p = 4.7 \cdot 10^{-8}$) and downregulated genes with those classified with repression memory ($p = 0.003$). Independently of any difference between wild-type and *rrp6Δ* strains in naive state, both strains can present transcriptional memory in response to galactose (Fig. 2B, Supplementary Fig. 2B).

Next, we focus on the potential ability of the nuclear exosome to modulate the observed transcriptional memory phenotype. To decrease strain-specific biases, we defined a *"transcription memory score"* (TM$_{score}$) for each strain and gene (Fig. 2D). Specifically, we compared the log$_2$ fold-change after 15 minutes after galactose addition in primed cells (t$_{15'}$) to 3 h after galactose addition (t$_{180}$) in naive cells (which is the maximum fold-change measured for most induced genes in naive cells). Comparing the transcriptional memory scores between wild-type and *rrp6Δ* strains, we identified 88 genes with enhanced activation memory in *rrp6Δ* (TM$_{score}$ *rrp6Δ* - TM$_{score}$ wild-type >0.58). As expected from our initial protein-based screen, we confirmed that *GAL1* displayed enhanced induction memory in *rrp6Δ* also in mRNA expression from its native locus. Induction memory genes enhanced by *RRP6* depletion also include other genes involved in aerobic respiration (Supplementary Fig. 2F and Supplementary Data 2). We identified 451 genes were *rrp6Δ* did not clearly affect the activation memory and 7 genes with attenuated activation memory in *rrp6Δ* (TM$_{score}$ *rrp6Δ* - TM$_{score}$ wild-type < −0.58). Next, we investigated if depletion of the nuclear exosome affected the dynamics of repression memory genes. We identified 158 genes with enhanced repression memory in *rrp6Δ* (TM$_{score}$ *rrp6Δ* - TM$_{score}$ wild-type < −0.58) including ribosomal protein and ribosome biogenesis genes (Supplementary Fig. 2G and Supplementary Data 2). We identified 611 genes where repression memory was not clearly impacted by RRP6 depletion and only 4 genes with attenuated repression memory (TM$_{score}$ *rrp6Δ* - TM$_{score}$ wild-type > 0.58). Interestingly, even if transcriptional memory measures controls for the expression of the naive cells, upregulated genes in the *rrp6Δ* strain overlapped with the genes classified with induction memory enhanced by *RRP6* depletion ($2.6 \cdot 10^{-8}$) and downregulated genes with those classified with repression memory enhanced by RRP6 depletion ($2 \cdot 10^{-7}$). Taken together, this shows that depletion of the nuclear exosome can enhance the transcriptional memory both for induced genes (making them increase their mRNA relative abundance faster) and for repressed genes (making them decrease their relative abundance faster).

## Non-coding RNA accumulation or chromatin reorganization do not explain transcriptional memory differences

After showing that multiple genes display an altered transcriptional memory in *rrp6Δ*, we investigated the potential mechanism by which the nuclear exosome could alter this process. Since accumulation of non-coding RNAs (ncRNAs) has been previously associated with a faster reactivation of *GAL* genes from repressive (glucose) conditions[20], we investigated if the observed differences in transcriptional memory could be explained by differential ncRNA accumulation. We focused on cryptic unstable transcripts (CUTs) that are well-known targets of the nuclear exosome and often arise bidirectionally from coding gene promoters[24,25]. First, we explored if the promoters of genes whose transcriptional memory was enhanced or attenuated after RRP6 depletion overlapped with previously annotated CUTs (*i.e.* – 150nt to −50 in the sense strand and −150 to +50 in antisense)[25]. However, we did not observe a clear association between CUTs overlapping promoters and transcriptional memory modulation by the nuclear exosome (Supplementary Fig. 3A). Focusing on the *GAL1*

region, we confirmed the expected gene expression remodeling in response to galactose treatment (Supplementary Fig. 3B). But we did not observe any clear accumulation of promoter-proximal ncRNA when comparing naive and primed states. To investigate the potential interaction between non-coding RNA accumulation and transcriptional memory beyond the GAL genes, we investigated the relative abundance of non-coding RNAs in naive and primed states (Supplementary Fig. 3C). However, we did not observe any clear global difference. Next, we used our transcriptomic data to explore potential differences in unannotated ncRNA transcription overlapping coding genes' promoters (sense and antisense) depending on their transcriptional memory classification (Fig. 3A and Supplementary Fig. 3D). Genes with induction memory enhanced after *RRP6* depletion displayed a clearly increased RNA-Seq coverage in the promoter region in the *rrp6Δ* strain in the sense strand in relation to the wild-type strain. However, no clear differences were observed between naive and primed states. To better study the difference between gene groups we defined the ratio t$_0$'/t$_0$ for each gene and compared the relative differences for each group (Supplementary Fig. 3E). Genes with repression memory had slightly lower non-coding transcription over the promoters in both the wild-type and *rrp6Δ* strain in comparison to repressed genes without memory. When investigating antisense transcripts covering promoter regions, we observed a general increase in RNA-Seq coverage for most groups in the *rrp6Δ* strain (Fig. 3A), as expected from antisense CUT accumulation[24,25]. Next, we investigated the differences in antisense transcription comparing the ratio t$_0$'/t$_0$ for each gene. We observed that genes with induction memory enhanced after *RRP6* depletion displayed an increased promoter antisense transcription in the wild-type strain, while that was not the case in the *rrp6Δ* strain (Supplementary Fig. 3F). As we observed only subtle differences between naive and primed conditions, we concluded that promoter-proximal ncRNA accumulation was likely not the main driver in the formation of transcriptional memory.

After studying the association between ncRNA accumulation and transcriptional memory, we investigated potential differences in chromatin status. It is well known that chromatin plays an important role in transcriptional memory[35], however, no detailed genome-wide investigation of chromatin organization comparing naive and primed cells has been conducted in this system. Thus, we performed high-resolution nucleosome mapping combining MNase treatment and chromatin immunoprecipitation in naive (t$_0$) and primed (t$_0$') cells for the wild-type and *rrp6Δ* strains. We did not observe any clear difference for global nucleosome occupancy comparing naive and primed states (Fig. 3B). We only observed a subtle displacement of nucleosomes −1 and +1 towards the nucleosome depleted region (NDR) in the *rrp6Δ* strain in both naive and primed cells. This *rrp6Δ*-specific alteration is reminiscent of the chromatin remodeling associated to an increase of antisense non-coding transcription that has been recently reported[36]. Next, we investigated nucleosome occupancy for induced and repressed genes (with or without memory), observing similar profiles (Supplementary Fig. 4A). As subtle changes in chromatin accessibility in *GAL1* promoter have been reported previously[33], we investigated also that locus specifically (Supplementary Fig. 4B) as well as a reference region containing known expressed and repressed genes as control (Supplementary Fig. 4C). However, we did not identify clear differences between naive and primed cells. To investigate subtle difference between gene groups we compared MNase coverage for t$_0$' and t$_0$ for the +1 nucleosome (0 to 150 bp from TSS) and for the Nucleosome Depleted region (NDR, −150 to 0 bp from TSS). The coverage for a +1 nucleosome increased for all analyzed genes group in primed conditions for the wild-type strain. For the *rrp6Δ* strain, MNase coverage was in general higher but without significant differences between naive and primed conditions (Supplementary Fig. 5A). As expected, MNase coverage in the NDR was lower across all regions (Supplementary Fig. 5B). To better study the difference between gene

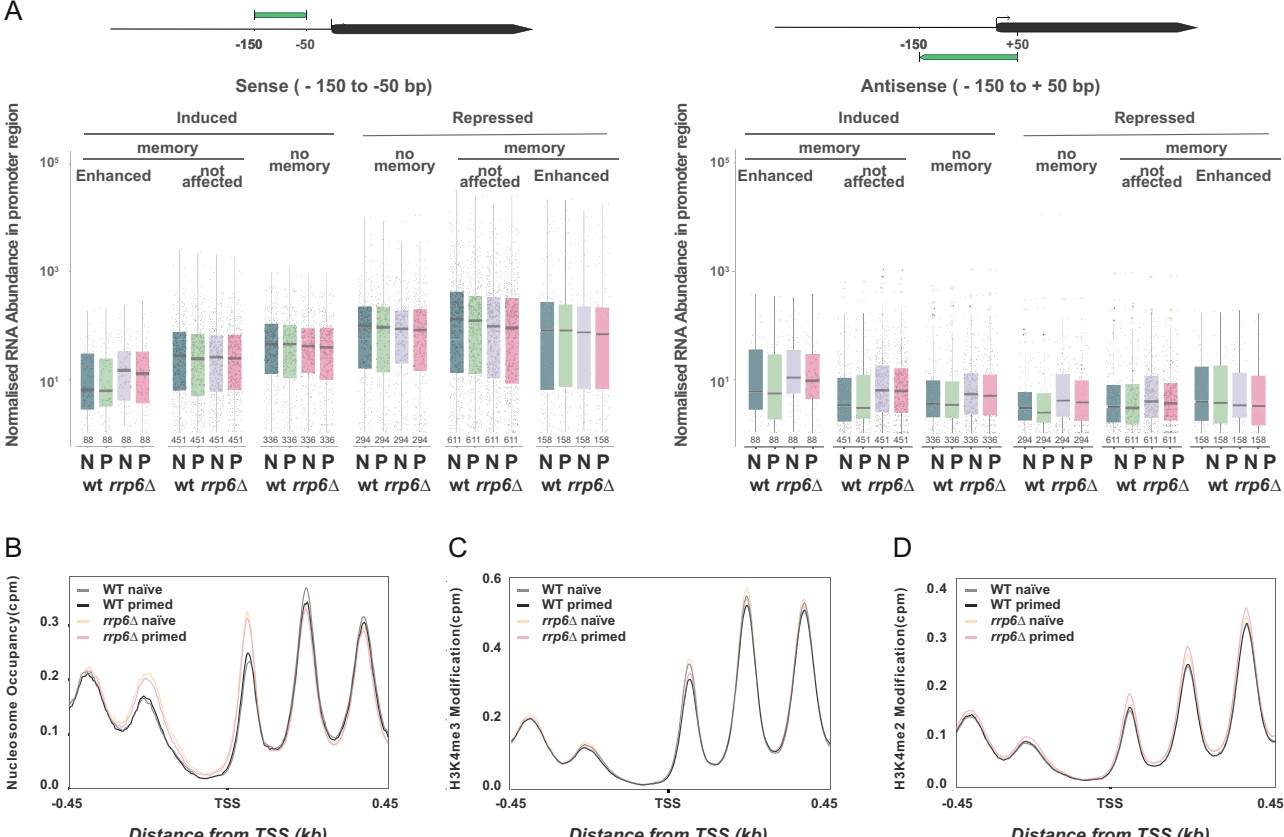

**Fig. 3 | Non-coding transcription and chromatin organization do not explain differences between naive and primed states. A** RNA-Seq coverage of unannotated ncRNA transcription surrounding coding genes' transcription start sites (TSS) (i.e., overlapping the promoter region). Average RNA coverage at the promoter region on the sense strand (−150 to −50 bp in respect to TSS) is shown on the left and antisense transcription (−150 to +50 bp with respect to TSS) is shown on the right. Boxplots for wild-type $t_0$ naive (N in cyan), wild-type $t_0'$ primed (P in green), $rrp6\Delta$ $t_0$ naive (N in purple), and $rrp6\Delta$ $t_0'$ primed (P in pink) are shown. mRNA abundance is normalized to global ORF-Ts abundance. Box-plot represents the first, second, and third quartile. The first quantile, median and third quantile are defined as the minimum, center, and maximum bounds of the box-plots. **B** Global MNase Seq analysis. Metagene plot of the distribution of average nucleosome MNase signal. Average sequencing coverage is shown (cpm, counts per million) for wild-type $t_0$ naive (grey), wild-type $t_0'$ primed (black), $rrp6\Delta$ $t_0$ naive (orange), and $rrp6\Delta$ $t_0'$ primed (pink) here. A 5 bp sliding window was used. **C** As B but for H3K4me3 ChIPseq signal. **D** As B but for H3K4me2 ChIPseq analysis.

groups we next defined the ratio $t_0'/t_0$ for each gene. We did not observe clear differences for $t_0'/t_0$ ratio for the +1 nucleosome across groups (Supplementary Fig. 5A). However, genes with repression memory enhanced by RRP6 depletion presented a clear decrease of NDR occupancy in primed conditions in the wild-type strain, but not in $rrp6\Delta$ strain.

In addition to potential changes in nucleosome occupancy, we also investigated histone marks previously associated to transcriptional memory. Even though in our experimental settings, we focus on short-term transcriptional memory to galactose (i.e., 3 h in glucose after initial galactose priming), we decided to explore potential changes in H3K4me3 and H3K4me2 that have been previously associated with long term transcriptional memory (>12 h) for galactose[18] and heat shock response in plants[35]. Comparing naive and primed cells at genome-wide level, we observed minimal differences in H3K4me3 and H3K4me2 (Fig. 3C, D). Although in some cases we identified some subtle differences between groups of genes and between strains (Supplementary Fig. 4A), we did not identify clear differences for +1 nucleosome or NDR coverage between naive and primed states (Supplementary Fig. 5C–F). However, when investigating difference between gene groups using gene-specific coverage ratio $t_0'/t_0$ we observed that genes with induction memory present relatively lower H3K4me2 in primed conditions in the $rrp6\Delta$ strain. In summary, although we observed subtle chromatin differences across groups, those differences alone

did not clearly explain the differential transcriptional memory observed in the $rrp6\Delta$ strain.

## Differential abundance of nuclear exosome co-factors modulates transcriptional memory

Having shown that chromatin-mediated regulation was not sufficient to explain the differences in transcriptional memory between wild-type and $rrp6\Delta$ strains, we focus on the effects of the nuclear exosome modulating mRNA stability. Although traditionally nuclear decay was thought to affect mainly non-coding transcripts[29], recent evidence shows that changes in nuclear RNA degradation rates can facilitate the remodeling of gene expression in yeast[30,31]. These changes can impact gene expression both in a positive and a negative way[30]. Specifically, in response to glucose deprivation, stress responsive genes can better escape nuclear decay (facilitating their accumulation). In contrast, genes downregulated in response to glucose withdrawal are targeted more efficiently by nuclear surveillance factors (facilitating their downregulation). Therefore, we hypothesized that the changes in nuclear decay observed in naive cells could be further enhanced in primed conditions and contribute to the observed $rrp6\Delta$-dependent transcriptional memory differences. To test the hypothesis, we investigated if genes with different transcriptional memory behavior present differential intrinsic affinity for the nuclear surveillance complexes TRAMP (Trf4/5-Air1/ 2-Mtr4-polyadenylation) or NNS (Nrd1-Nab3- Sen1). We used previously published CRAC (crosslinking

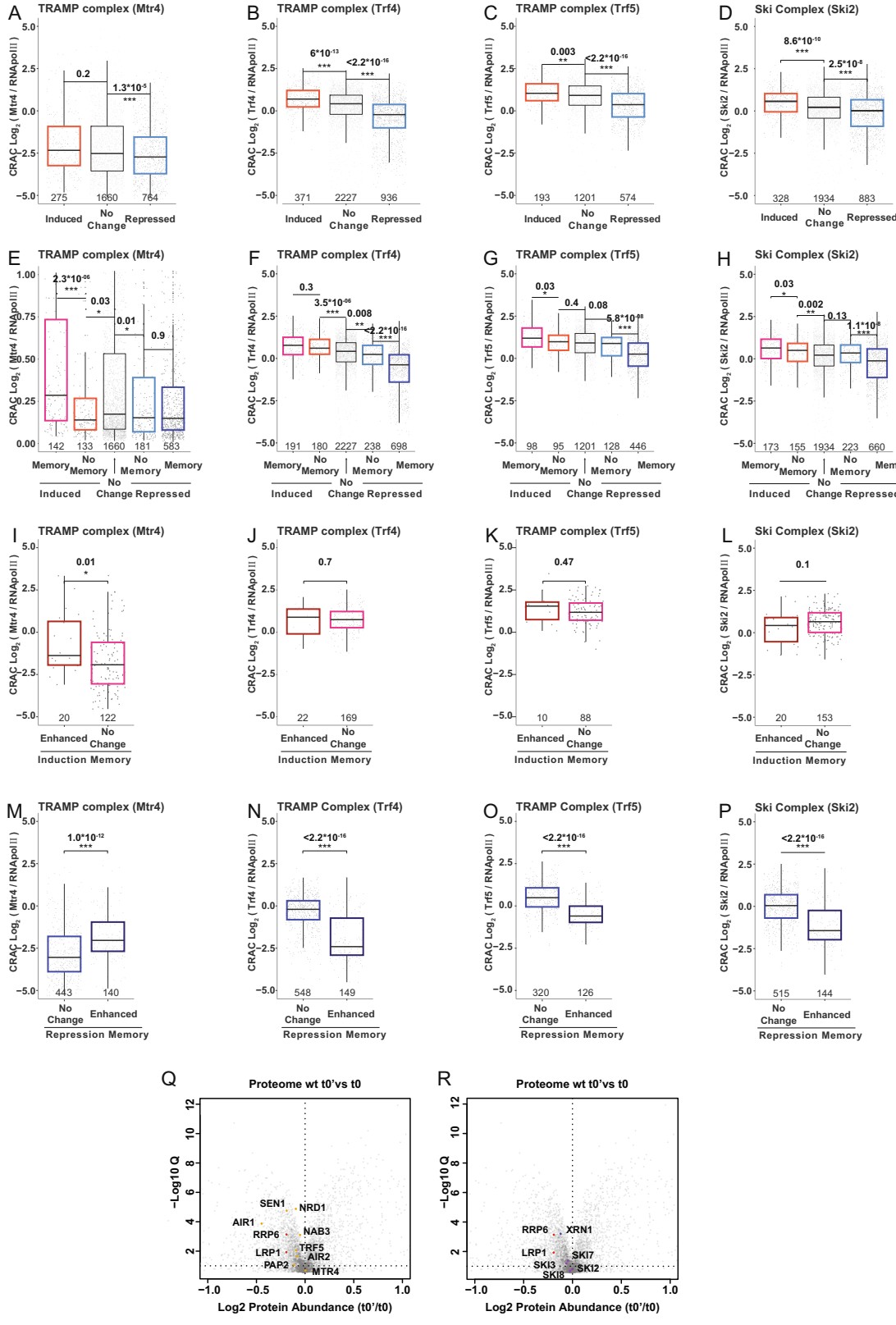

and cDNA analysis) data obtained in naive conditions measuring the intrinsic association of specific mRNAs to the TRAMP[30,37] (i.e. Mtr4, Trf4, Trf5) NNS[30] (i.e. Nab3) and the SKI complex[38] (i.e., Ski2) normalized by their binding to RNA Polymerase II (Supplementary Data 3). We observed that genes induced after galactose addition generally exhibit higher intrinsic association to both complexes, while genes repressed in response to galactose are relatively less bound (Fig. 4A–D and

Supplementary Fig. 6A). Interestingly, this trend was even more clear when analysing genes with induction and repression memory. Specifically, induced genes with transcriptional memory (for which mRNA abundance increases faster in primed cells) are more associated to TRAMP and NNS complexes, while repressed genes with transcriptional memory are less associated (Fig. 4E-H and Supplementary Fig. 6B). Next, we explored up to what degree differential intrinsic

**Fig. 4 | Differential association to nuclear exosome co-factors. A** Relative association for the TRAMP complex (i.e. Mtr4) as measured by CRAC in naive cells[30]. Boxplot for induced and repressed genes is shown. To measure gene-specific nuclear surveillance association, since those complexes act on nascent transcripts, CRAC data is normalized by RNA pol II association, as previously described. Number of analyzed genes is indicated in grey. Genes with less than 20 counts were discarded. First quantile, median and third quantile are defined as the minimum, center and maximum bounds of the box-plots. Two-sided Wilcoxon test was used. **B** as A but measuring the relative association of Trf4 (Pap2)[37] **C** as A but measuring the relative association of Trf5[37]. **D** as A but measuring the relative association for the Ski complex (Ski2)[38]. **E–H** as in A-D but for genes classified according to their transcriptional memory. **I–L** As in A-E but comparing genes with induction memory enhanced or not enhanced by *RRP6* depletion. **M–P** As in A-D but comparing genes with repression memory enhanced or not enhanced by RRP6 depletion. Significance computed using Wilcoxon signed-rank test. CRAC data from Bresson et al.[30] Tuck et al.[38] and Delan-Forino et al.[37] Trf5, TR5 and Ski2 data was normalized using RNApol II CAC data from Bresson et al.[30] **Q** Volcano plot showing relative protein abundance changes of primed (t0′) and naive (t0) states in wild-type cells. Fold changes in protein abundance for t0′ and t0 samples are shown as a function of statistical significance ($n = 4$ of independent biological replicates). The points showing the protein subunits of the TRAMP, NNS, and nuclear exosome are labeled. Doted lines represent $Log_2$ fold change = 0 and $-Log_{10}Q = 1$. **R** Same data shown in Q with labels pinpointing components of the nuclear and cytoplasmic exosome and the 5′–3′ exonuclease XRN1.

association to the nuclear surveillance complexes could explain the transcriptional memory phenotype observed in the *rrp6Δ* strain. We observed that genes, where induction memory was enhanced by the nuclear exosome depletion, had in general higher association to TRAMP and NNS (Fig. 4I-L and Supplementary Fig. 6C). On the contrary, repressed genes whose repression memory was enhanced by the nuclear exosome depletion had in general lower association to NNS (Fig. 4M–P and Supplementary Fig. 6D). Interestingly, Mtr4 presented an opposite behavior in respect to the other TRAMP components (Trf4/5) suggesting a differential role in repression memory. Taking all the CRAC data together, our analysis suggests that differential affinity to nuclear surveillance factors contribute to transcription memory.

The differential association between an RNA binding protein (RBP) and their mRNA targets can be caused by two alternatives (and not exclusive) mechanism: 1) by changes in intrinsic mRNA-RBP affinity, or 2) by changes in RBP abundance. To test this second possibility, we performed a proteomic analysis of wild-type cells in naive and primed conditions (Fig. 4Q and Supplementary Data 4). This analysis revealed a clear decrease of the components of the TRAMP, NNS and nuclear exosome components in primed conditions. The decrease of components involved in nuclear decay would be consistent with a relative increase in expression for those genes more sensitive to its action (*e.g.*, genes displaying induction memory). Unexpectedly, we also observed a relative decrease of factors involved in cytoplasmic decay, although to a lower degree than the nuclear ones (Fig. 4R). However, that was not so clear in the *rrp6Δ* strain (Supplementary Fig. 6E, F). This suggests that changes in the cytoplasmic mRNA decay may also contribute to modulate transcriptional memory.

Finally, we took advantage of the generated proteomic data to investigate the potential role of Gal1p cytoplasmic accumulation in transcriptional memory. It has been shown that cytoplasmic accumulation of Gal1p can facilitate faster long-term transcriptional memory in primed cells[21,22]. Consistent with that, we observed that Gal1p was reliably detected in primed cells (Supplementary Fig. 6G). However, both the wild-type and the *rrp6Δ* strain present similar levels for Gal1p suggesting that its cytoplasmic accumulation does not explain the observed differences in transcriptional memory between strains.

## Changes in cytoplasmic RNA stability also modulate transcriptional memory

Having investigated the role of nuclear RNA degradation in transcriptional memory, we decided to study the potential role of other RNA degradation pathways. We reasoned that, as targeting for nuclear RNA degradation is often set co-transcriptionally[28] changes in nuclear mRNA stability will mainly modulate the appearance of newly synthetized molecules. However, it could be expected that, to modulate the abundance of mRNAs already presents in the cytoplasm (e.g. genes with repression memory), cytoplasmic mRNA decay would be also altered between naive and primed cells. Unfortunately, key components of the cytoplasmic mRNA decay such as *XRN1* lacked sufficient coverage to be included in our analysis (Supplementary Data 1). To investigate if global changes in mRNA stability (and not only nuclear

decay) contribute to transcriptional memory we measured genome-wide mRNA stability using metabolic RNA labeling (SLAM-Seq)[39,40] in naive and primed cells. SLAM-Seq compares total mRNA abundance to that of new mRNA molecules generated during the metabolic RNA labeling pulse. Thus, it can be expected that it captures mainly changes in mRNA stability due to differences in cytoplasmic mRNA decay, while changes in nuclear decay would be difficult to measure (as nuclear RNA decay acts mainly on nascent mRNA molecules). We labeled newly synthetized RNA with thiouracil for 10 min and harvested cells at 0 and 30 min after galactose addition for both wild-type and *rrp6Δ* strains (see methods). As expected upon shift to galactose, the generation of newly synthesized mRNA molecules was drastically decreased[41] (Supplementary Fig. 7A). Thus, we focus our analysis on naive ($t_0$) and primed ($t_0'$) conditions where we could assume a steady state between mRNA synthesis and decay (i.e. synthesis rate is in equilibrium with the mRNA decay rate) and control for potential differences in the response to galactose between naive and primed states.

Investigating the wild-type strain, we observed a clear decrease in mRNA turnover in the primed condition, indicating a general stabilization of the cellular mRNAs ($p$-value $< 2.2 \cdot 10^{-16}$, Fig. 5A). Although most mRNAs increase their mRNA stability in primed cells (slower turnover), genes associated to respiration and galactose metabolism were particularly stabilized (Supplementary Fig. 7B and Supplementary Data 5). On the other hand, genes associated with cytoplasmic translation were less stabilized (Supplementary Fig. 7C and Supplementary Data 5). Next, we investigated the changes in mRNA turnover between naive and primed conditions for different groups of genes. Focusing on genes induced after galactose treatment, we observed that their mRNA stability was significantly increased (slower turnover) in primed cells (Fig. 5B), something that could facilitate their accumulation upon re-induction. The stability of induced genes with transcriptional memory was increased even more (Fig. 5C and Supplementary Fig. 7D, $p$-value $< 2.8 \times 10^{-6}$). On the contrary, the mRNA stability of genes repressed after galactose treatment increased less than the global population (i.e., less decrease in mRNA turn-over, Fig. 5B). And again, the trend was even more clear for mRNAs with repression memory (Fig. 5C and Supplementary Fig. 7D). In agreement with our model, maintaining a relatively faster turnover (lower mRNA stability) in primed cells could contribute to the fast decrease of their mRNA abundance upon transcription shut down. Next, we investigated how the stability of repression memory genes enhanced by *RRP6* depletion, changed between naive and primed conditions. We observed a decrease in stability in respect to other repression memory genes (Fig. 5D), suggesting that changes in mRNA stability contribute to the observed transcriptional memory phenotype. Genes with induction memory enhanced by *RRP6* depletion presented a subtle relative increase in mRNA stability in primed cells, but this change was not significant (Supplementary Fig. 7E). This could be caused by a different regulation mechanism or due to our limited ability to accurately measure their mRNA turnover in non-induced conditions (where their mRNA abundance is extremely low). To confirm that mRNA stability differences measured between naive and primed conditions, was

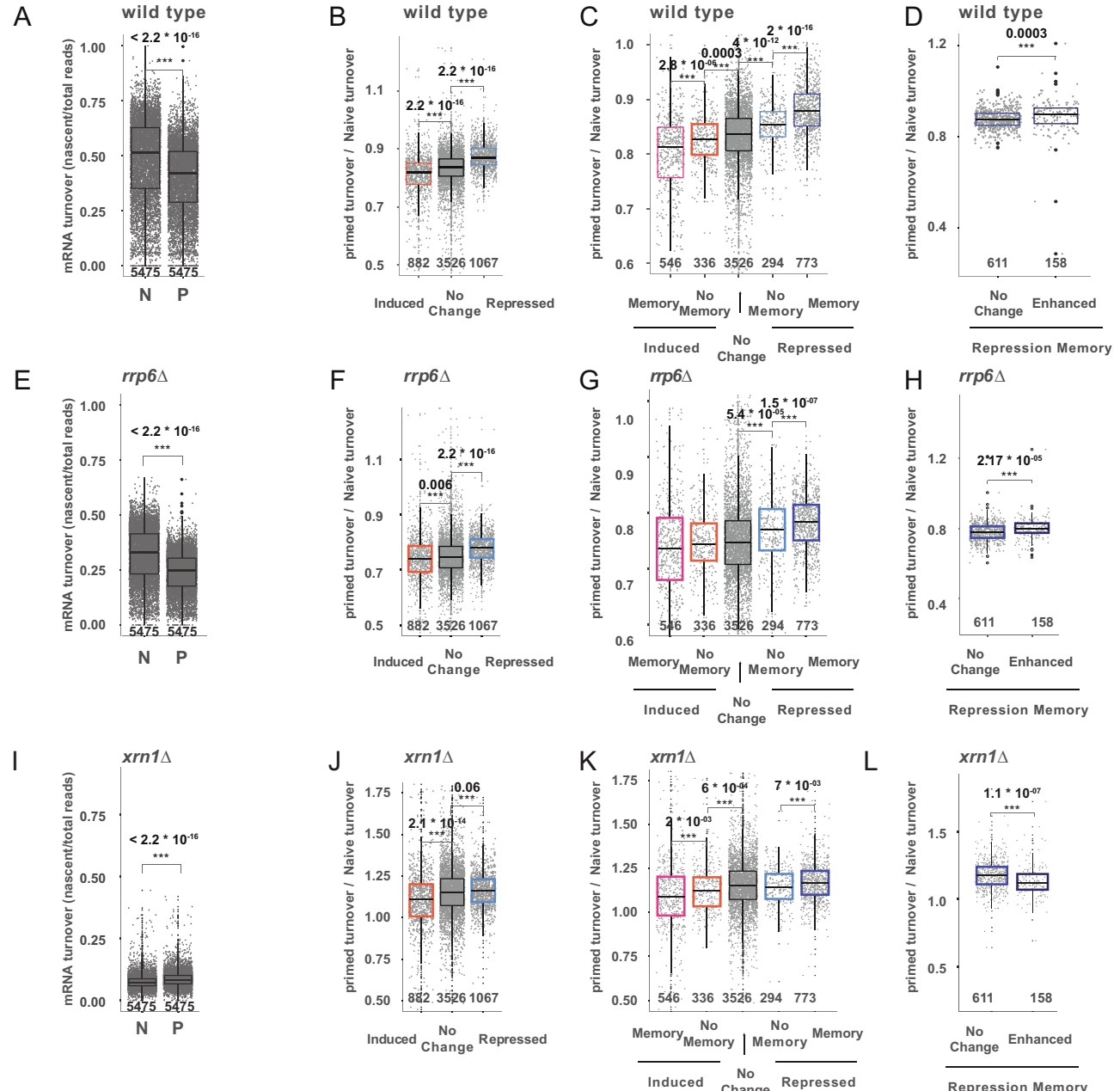

**Fig. 5 | Differential mRNA turnover between naive and primed cells. A** Relative mRNA turnover (comparing nascent vs total RNA) using SLAM-seq in naive (N, $t_0$) and primed (P, $t_0'$) conditions. High turnover indicates faster transcription/decay and thus lower mRNA stability. Gene-specific turnover was computed by comparing for each gene the reads containing T > C conversion to the total mapped reads. Only reads containing at least 2 T > C conversions are considered newly synthesized reads. Only genes with at least 20 total reads were considered for RNA turnover analysis. **B** Change in mRNA turnover between primed and naive conditions for induced and repressed genes in response to galactose. Number of analyzed genes is shown at the bottom of each boxplot. **C** As in B but for genes according to their transcriptional memory profile. **D**, As in B but for genes with repression memory enhanced or not enhanced after RRP6 depletion. **E–H** As A-D but for the *rrp6Δ* strain. **I–L** As A-D but for the *xrn1Δ* strain. For all boxplots the number of genes examined is depicted at the bottom of the boxplot and the independent biological experiments $n = 3$. First quantile, median and third quantile are defined as the minimum, center and maximum bounds of the box-plots. Two-sided Wilcoxon test was used for all comparisons.

also present during the galactose response we measured mRNA stability using a pulse-chase approach. We labeled RNA for 1 h in naive and primed conditions and measured its disappearance upon transition to galactose for 30 minutes (Supplementary Fig. 7F). This revealed changes in mRNA turnover similar to the group-specific regulation described using a steady-state approach in YPD (Supplementary Fig. 7G–J).

Next, we expanded our analysis to investigate global changes in mRNA stability in the *rrp6Δ* strain. Using SLAM-Seq, we determined that *rrp6Δ* has a slower mRNA turnover than the wild-type strain

(Fig. 5E, *p*-value < $2.2 \times 10^{-16}$). As expected, this effect was particularly strong for CUTs, which were clearly stabilized in *rrp6Δ* (Supplementary Fig. 8A, *p*-value < $2.2 \times 10^{-16}$). Having validated our RNA turnover measurements in *rrp6Δ*, we investigated differences in mRNA turnover between naive and primed conditions. As in the wild-type strain, we observed a clear decrease in mRNA turnover in primed *rrp6Δ* cells (Fig. 5E, Supplementary Fig. 7A, *p* < $2.2 \times 10^{-16}$). This shows that the observed mRNA stabilization in primed cells (decreased mRNA turnover) does not depend exclusively on the nuclear exosome, but that regulation of the cytoplasmic mRNA decay is also involved. In

agreement with that observation, RNA turnover regulation between naive and primed conditions was similar to that of the wild-type strain. Specifically, genes displaying induction memory underwent a significant increase in their relative mRNA stability (slower turnover) in primed cells, while genes with repression memory decreased their relative RNA stability (faster turnover) in primed cells (Fig. 5F–H, Supplementary Fig. 8A–C).

To further dissect the contribution of cytoplasmic factors to this process, we investigate global changes in mRNA turnover in *ski2Δ* (component of the cytoplasmic 3'-5'exosome) and *xrn1Δ* (cytoplasmic 5'-3'exonuclease) strains. When investigating the *ski2Δ* strain, we also observed a general stabilization of mRNAs in primed cell (Supplementary Fig. 8D). Like in the wild-type strain, genes displaying induction memory increased in their relative mRNA stability (slower turnover) in primed cells, while genes with repression memory decreased their relative RNA stability (faster turnover) in primed cells (Supplementary Fig. 8D–I). Finally, we investigated mRNA turnover in *xrn1Δ* cells. As expected, deletion of XRN1 led to a massive mRNA stabilization (Fig. 5I), but contrary to in the previous cases, mRNA in primed cells was less stable than in naive conditions ($p < 2.2 \times 10^{-16}$). However, despite this lack of stabilization of mRNA in primed conditions, when comparing gene-specific differences between naive and primed conditions we observed the same group-specific relative changes in mRNA turnover as for the other strains (i.e., induction memory genes increasing their relative mRNA stability in primed cells and repression memory decreasing it) (Fig. 5J–L and Supplementary Fig. 8J, K). This shows that cytoplasmic 5'-3'mRNA decay is not essential for most gene-specific effects, and rather has a role on global stability change between naive and primed states. However, and in contrast to what we observed in the other strains, genes with repression memory enhanced by *RRP6* depletion presented a relative decrease in their mRNA stability in primed cells compared to other repression memory genes (Fig. 5P, $p < 1.123 \times 10^{-7}$). Interestingly those same genes present a particularly high codon adaptation index (Supplementary Fig. 8L). As codon optimality has been shown to be a main player controlling cytoplasmic mRNA stability[42], our results suggest that those genes are more dependent on the action of XRN1, and potentially codon optimality, to regulate their stability between naive and primed conditions. Taken together, our results show that changes in mRNA stability between naive and primed states contribute to the transcriptional memory phenotype. And that both nuclear and cytoplasmic mRNA degradation contributes to this process.

## Discussion

Previous studies investigating transcriptional memory have focused on how direct modulation of the transcription process enables faster gene re-induction kinetics. However, even though steady-state mRNA levels reflect the combination of transcription and RNA decay, the role of mRNA degradation in this process remained unexplored. Here we show that differential regulation of mRNA stability facilitates gene expression adaptation in response to environmental changes and that this process can be influenced by previous stimuli.

In this study, we performed a genome-wide screen to identify novel factors able to modulate transcriptional memory in budding yeast in response to galactose as a carbon source. In addition to confirming the role of known players in this process, we show that the depletion of the nuclear exosome leads to faster reinduction kinetics of the *GAL1* gene in galactose-primed cells. To dissect this process, we performed a detailed RNA-Seq study of gene expression changes in naive and primed cells. Consistent with previous work[34], we identify that transcriptional memory both facilitates the induction of particular genes (activation memory) and also accelerates the repression of others (repression memory). Unexpectedly, when investigating the role of the nuclear exosome in this process, we discovered that its depletion could enhance both activation and repression memory.

Specifically, RRP6 depletion enhanced the transcriptional memory of a subset of induced genes (making them increase their mRNA abundance faster) and also the repression of a different subset of genes (making them decrease mRNA abundance faster). The groups displaying enhanced transcriptional memory phenotypes were not random. For example, genes related with carbohydrate metabolism or meiosis displayed enhanced activation memory in *rrp6Δ*, while cytoplasmic translation was enriched in the group of genes with enhanced repression memory (Supplementary Fig. 2 and Supplementary Data 2). As the nuclear exosome has a well-known role in controlling promoter directionality, transcription termination and the abundance of non-coding transcripts (i.e., CUTs), we first investigated its potential role in controlling the transcription process. However, our analysis suggests that the accumulation of CUTs overlapping promoters of coding genes[5] are likely not the main mechanism of action controlling those differences. Likewise, we also discarded a major role of potential chromatin differences when comparing naive and primed states.

Next, as our result showed that the depletion of the nuclear exosome led to divergent effects in groups of genes that are regulated in opposite directions during cellular adaptation, we considered if the effect could be mediated by regulation of mRNA decay (instead of changes in the transcription process). Recent work shows that the nuclear exosome, in addition to its canonical role in regulation of ncRNA abundance, is also important in facilitating remodeling of gene expression in yeast[30,31]. With those works as a starting point, we investigated if, in addition to its role in modulating coding mRNA abundance during the stress response, the nuclear exosome could also behave differently in naive and primed conditions. Our analysis showed that genes whose induction memory was enhanced by the nuclear exosome depletion had in general higher association to nuclear surveillance factors (*i.e.*, NNS and TRAMP complexes). On the contrary, repressed genes with memory enhanced by the nuclear exosome depletion had in general lower intrinsic association to these factors. This suggested a scenario where the nuclear exosome could limit the accumulation of mRNAs from genes with activation memory, while having almost no effect in mRNAs from genes with repression memory. This would explain our observations, since in absence of the nuclear exosome, mRNAs from genes with activation memory would accumulate faster. On the contrary, nascent mRNAs from genes with repression memory would not be particularly stabilized in absence of the nuclear exosome. However, as most mRNAs would undergo a subtle stabilization in absence of the nuclear exosome, this would mean that mRNAs from genes with repression memory would decrease in their relative abundance faster than the global population. As our working model required a change in the activity of the RNA degradation machinery in primed cells, next we used MS-based proteomics to compare naive and primed cells. Reassuringly, we found that primed cells present a relative lower abundance of proteins involved in nuclear mRNA decay (Fig. 4U).

Finally, since the contribution of nuclear decay can be expected to be relatively small in comparison to cytoplasmic decay, we investigated if global changes in mRNA stability (and not only nuclear decay) could contribute to differences between naive and primed cells. We reasoned that changes in cytoplasmic mRNA stability would be especially important in facilitating a faster downregulation of mRNAs of genes with repression memory, as in those genes transcriptional activity, and consequently nascent nuclear mRNA decay, can be expected to be low. To test this hypothesis, we measured RNA stability in primed and naive cells using RNA metabolic labeling. Despite naive and primed cells displaying an almost identical gene expression program (Fig. 2, Supplementary Fig. 2), our results clearly show that primed cells undergo a decrease in mRNA turnover, indicating a general stabilization of the cellular mRNAs (Fig. 5A). However, this stabilization is not equal across all gene groups. In fact, genes

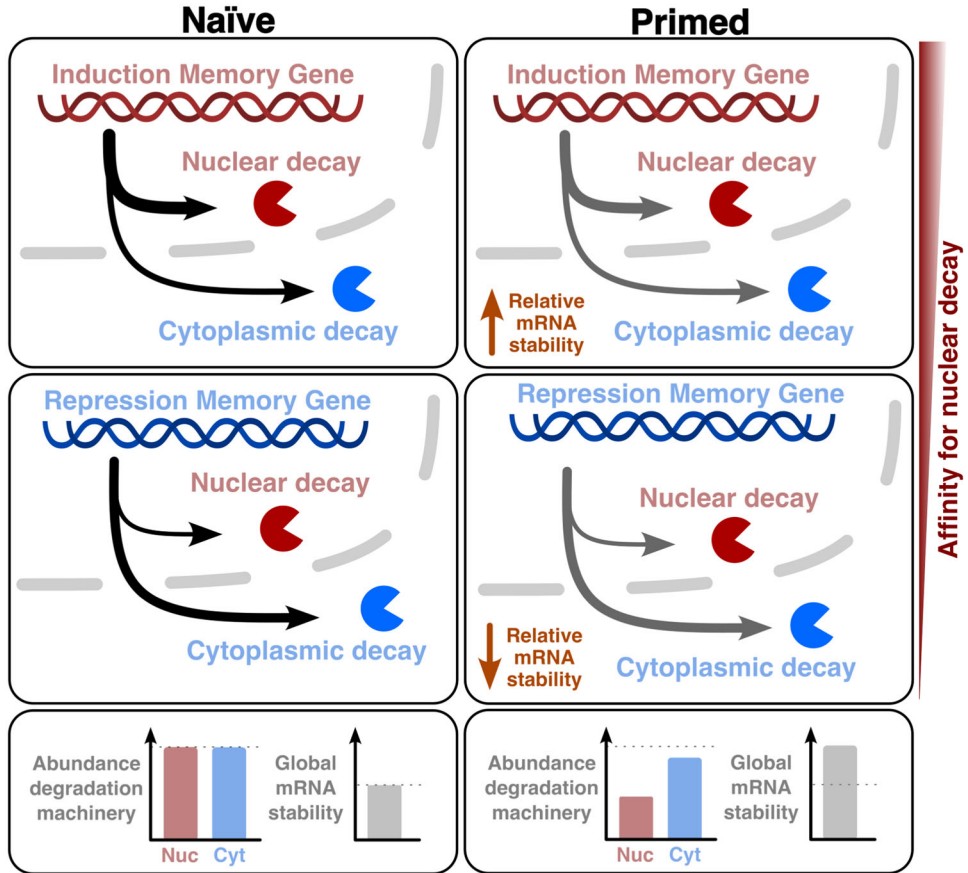

**Fig. 6 | Working model for the role of mRNA stability in transcriptional memory.** Induction memory and repression memory genes present different intrinsic susceptibilities to nuclear mRNA decay. After galactose priming, the abundance of nuclear degradation machinery decreases. Thus, although there is a global increase in absolute mRNA stability, the relative stability of induction memory genes increases and the one of the repression memory genes decreases. These favors enhanced induction/repression responses in primed cells.

that are induced after galactose treatment exhibit significantly higher mRNA stability in primed cells, something that would facilitate their accumulation upon re-induction. This stabilization was even more pronounced for genes with activation memory. On the contrary, genes that are repressed after galactose treatment exhibit a lower relative increase in their mRNA stability in primed cells in respect to the global population. This relative destabilization was even more clear for genes with repression memory, something that would facilitate their faster decrease upon galactose re-stimulation. Interestingly, although our work shows involvement of RRP6 in regulation of stability in a subset of repression memory genes, we also observe clear changes in mRNA stability between primed and naive cells even in the absence of the nuclear exosome. To further dissect the role of other degradation pathways we measured mRNA turnover in a deletion strain for a component of the cytoplasmic 3′-5′exosome (*ski2Δ*) and for the major cytoplasmic 5′-3′exonuclease (*xrn1Δ*). In the case of the *ski2Δ* strain we observed similar results as in the *rrp6Δ* strain. However, in the *xrn1Δ* strain the general stabilization of mRNAs in primed conditions was lost (Fig. 5I). This suggests that the differential activity for XRN1, considered to be the main degradation pathway, is essential to achieve the global differences in mRNA stability between naive and prime conditions. Reassuringly, the general changes observed in mRNA turnover in the *xrn1Δ* strain are consistent with the relative decrease of Xrn1p abundance that we observed in primed conditions in the wild-type strain. However, we did not capture XRN1 in our initial screen as that strain was below our detection limit

(Supplementary Data 1). Despite the clear role of XRN1 controlling global mRNA stability, the relative changes in mRNA turnover for groups with differential transcription memory between naive and primed conditions was largely maintained (Fig. 5J-L). Interestingly those genes with higher relative destabilization in primed conditions present a particularly high codon adaptation index (Supplementary Fig. 8H) and are differentially regulated in the *xrn1Δ* strain (Fig. 5L). This suggest that they could be more sensitive to drastic changes in translation (and thus co-translationally regulated cytoplasmic mRNA stability). However, more research would be required to investigate this possibility.

Taking all this together, our work suggests that there is a co-regulation between the transcriptional responses associated to faster transcriptional induction and repression and the associated changes in post-transcriptional mRNA life (at nuclear and cytoplasmic levels) (Fig. 6). This suggests a general connection between transcription rate and nuclear and cytoplasmic mRNA decay, which together modulate mRNA abundance. Thus, our work suggests the existence of coordination between "*transcriptional*" and "*post-transcriptional memory*" to facilitate swifter adaptation to changing environments. Although here we have focused on the dissection of gene expression during carbon-source change in budding yeast genes, we anticipate that similar synergistic transcription-mRNA stability crosstalk could occur in other conditions where massive gene expression changes are expected. Understanding how transcription activity and mRNA decay can change while maintaining apparently identical mRNA abundance will be important to understand differential behavior in adaptation to changing environments and cellular response.

## Methods

### Generation of a genome-wide reported for pGAL1 transcriptional memory

To generate the pGAL1 memory reporter system we used p416 TEF as backbone[43]. The final plasmid contains a constitutive reporter with a pTEF promoter expressing mCherry-degron and a CYC1 terminator. The galactose reporter is controlled by a pGAL1 promoter expressing sfGFP-degron and an ADH1 terminator. The plasmid confers Nourseothricin resistance (NAT1) to enable its transformation into the barcoded yeast deletion collection containing KanMX[32] (Supplementary Fig. 1A). To obtain repeated measures of a wild-type strain in response to glucose-galactose transition we generated 8 barcoded control strains where we introduced a KanMX cassette containing strain-specific barcode next to the $ura3\Delta O$ locus in a BY4741 strain. We transformed the diploid deletion[32] collection with the pGAL1 memory reporter plasmid using Gietz's Frozen competent yeast transformation protocol[44]. The pooled collection was grown overnight in YPD ( + 200 μg/ml Geneticin) and processed in exponential phase $OD_{600}$ ~ 0.5. To increase sample complexity and minimize cell bottlenecks we performed multiple transformations. For each pool transformation, we used $10^8$ cells and 236 μg plasmid DNA. The heat shock was performed for 1 h at 42 °C in a heat block with 5 min 1000 rpm interval shacking and additional vertexing every 20 min. Antibiotic selection was performed on Nourseothricin (60 μg/ml) plates to minimize competition between adjacent clones. We performed a total of 15 independent transformations that led to ~30,000 clonal colonies. Colonies were scratched, mixed with glycerol (final 21%), aliquoted and frozen. For the 8 barcoded control strains we performed the same process but transforming independently each strain.

### Reporter based screen for transcriptional memory

Frozen aliquots of the barcoded genome-wide deletion collection barcoded controls containing the reporter system were revived for 8.5–10 h in YPD + Nourseothricin (60 μg/ml). We performed in parallel the experiment with the pooled genome-wide deletion collection and the pooled barcoded controls. When cells reached exponential growth ($OD_{600}$ 0.4–0.6), we took and aliquot and defined that point as TP0 (naive). We changed media to galactose (YPGal) for 3 h and collected an aliquot every hour (TP1 = $t_{60}$, TP2 = $t_{120}$, TP3 = $t_{180}$). When then changed cells back to glucose media for 3 h and collected TP4 ($t_{90GLU}$) and TP5 ($t_{180GLU}$ which served also as $t_0'$). Finally, we re-exposed cells to galactose for 3 h and collected a sample each hour (TP6 = $t_{60}'$, TP7 = $t_{120}'$, TP8 = $t_{180}'$). Media change was performed by spinning cells (5 min 2500 g) and performing one wash with the new media before final resuspension. For fluorescent detection cells were fixed in 50% ethanol (final) and stored at 4 °C. Fixed pooled barcoded controls and pooled genome-wide deletion were combined at 1:65.5 cell ratio (every barcoded control would correspond to 1:500 of all cells). Before fluorescent measurement, cells were spun down (5 minutes 3000 g), washed with PBS and spun down for 30 minutes to allow the sfGFP to refold. After this cell were resuspended in PBS for FACS.

For FACS sorting we fixed gates to consider only those cells significantly expressing MCherry. After this filter, we split sfGFP expression in 4 gates: no expression (R5), low expression (R6), intermediate expression (R7) and high level of sfGFP expression (R8). We aimed to sort a minimum of $10^6$ cells per gate when feasible, and sorted between 7 and 25 million cells per time point (see DataS2 for details). We performed two independent replicates.

For DNA isolation cells were resuspend in 300 μL of 10'prep buffer (2% TritonX-100, 1% SDS, 100 mM NaCl, 1 mM EDTA y 10 mM Tris-HCl pH 8) and transferred to a 2 mL tube with 500 μL Phenol:Chlorophorm:isoamilic (25:24:1) and 500 μL glass beads. Cells were vortexed using 4 30 second pulses in a FastPrep-24 (MP Biomedical) instrument at 5.5 m/s. We recovered the aqueous phase after centrifugation and a gel-lock tube with 500 μL

Phenol:Chlorophorm:isoamilic (25:24:1). We vortexed the sample, centrifuge and recovered the aqueous phase. We added 3 L RNase A (DNase-free) (10 mg/mL) and incubate during 30 min at 37 °C. Ilumina compatible sequencing libraries of the UpTag region of each strain were generated by two consecutives PCRs. During the first PCR we used oligos indexing for the biological replicate (KanMX) and the time points (UPTAG). During the second PCR we introduced illumine-compatible oligos and indexed for sGFP signal (see DataS2 for details). We performed 4 independent PCRs for each sample. In brief, for the first PCR reaction contained 2 μL extracted gDNA, 1 μl dNTPs (10 mM), 0.25 μl UpKanMX primers (10 mM), 0.25 μl UPTAG primers (10 mM), 10 μl 5 x Phire Reaction Buffer, 0.3 μl Phire Hot Start II DNA Polymerase (Thermo Fisher Scientific) in a total reaction volume of 50 μl. PCR program was conducted for 30 s at 98 °C, 15–16 cycles of 10 s at 98 °C, 10 s at 63 °C, 30 s at 72 °C and final elongation for 5 min at 72 °C. For the second PCR we used 2 μl product from the first PCR, 1 μl dNTPs (10 mM), 0.5 μl PE2_MPX (10 mM), 0.5 μl PE1.0 (10 mM), 10 μl 5 x Phire Reaction Buffer, 0.3 μl Phire Hot Start II DNA Polymerase (Thermo Fisher Scientific) in a total reaction volume of 50 μl. PCR program was conducted for 30 s at 98 °C, 15-16 cycles of 10 s at 98 °C, 10 s at 65 °C, 30 s at 72 °C and final elongation for 5 min at 72 °C. PCR replicates were pooled, purified with HighPrep beads (MagBio), quantified and pooled at equal concentration. Libraries were sequenced using the Illumina HiSeq2000 platform.

### Transcriptional memory screen analysis

To assign mapping sequenced barcodes onto strains we constructed a fasta contig for each deletion strain with its uptag and downtag, and these fasta contigs were indexed using bwa. The sequencing reads were trimmed down to the first 45 bp and then aligned using bwa aln (default settings with −B 6) and bwa se (default settings with −n 1). The first 6 bp of the barcode that indicates the timepoint of the sample was placed under the BC tag during alignment with bwa (the −B 6 option). After alignments, counts for each deletion strain in each timepoint and window were tabulated using R.

We focus on the comparison of the differences between GFP accumulation in naive (TP3-4) and primed (TP5-6) cells. Please note that, as protein accumulation is delayed with respect to mRNA production, TP4 corresponds to cells already transitioned back to glucose.

To test for differences in memory response we modeled the change in barcode counts for each strain using a quasibinomial generalized linear model. For each transition (e.g TP3 to TP4), for timepoint i, biological replicate j and strain k

$$\log(Counts_{i,j,k = k}/Counts_{i,j,k = wildtype,}) \sim biological\_replicate + timepoint$$

(1)

The extra dispersion parameter for quasibinomial glm is estimated from the data using Pearson's coefficient of dispersion (implement by quasibinomial function in R)

For transition TP3 to TP4, we can obtain an estimate of the change in log odds ratio between TP4 and TP3, $\beta_1$ and also an estimate of its error, $se_1$. Likewise in transition from TP5 to TP6, we obtain an estimate $\beta_2$ and $se_2$. To ask if the responses during the second transition (TP5 to TP6) is significantly different from the first transition (TP3 to TP4), we calculate a z-score for this:

$$Z = (\beta_1 - \beta_2)/sqrt(se_1 + se_2)$$

(2)

Then we combined the z scores for 3 windows using Stouffer's method, using the cell counts in each windows as weights. It is to be noted that we take the absolute of all Z values since they can take different signs. After calculating this estimate of Stouffer Z, we check whether the z scores change signs more than once before calling it a significant change in responses. We identify 35 mutants with putative

decreased transcriptional memory (TP5_6 < TP3_4) and 37 with enhanced transcriptional memory (TP5_6 > TP3_4) (Supplementary Data 1)

To calculate GFP scores of individual strains at the different timepoints, several data manipulation were performed. First, a cutoff was applied to only include abundant strains (rowMeans > 5). To obtain the relative abundance of the strain in the corresponding sample, for each sample, reads of individual strains were normalized by dividing them with the total number of their samples. Wild type and deletion strains have shown a dispersed distribution across the 4 GFP windows and across time points. In order to obtain the averaged GFP expression of individual strains at the different time points, the contribution of each of 4 GFP windows to average GFP expression has to be normalized. Thus, relative numbers of FACS sorting events for the 3(4) different GFP fractions per time point were multiplied with the corresponding relative reads of individual strains at the different time points. Next for each GFP window a score was assigned ($GFP^- = 1$, $GFP^{low} = 2$, $GFP^{mid} = 3$, $GFP^{high} = 4$). Normalized reads were multiplied with their respective GFP score, summed up for individual strains at individual time points and divided by sum of the non-multiplied values.

## RNA-Seq experiment and library preparation

*Saccharomyces cerevisiae* strain BY4741 (MAT a *his3Δ1 leu2Δ0 met15Δ0 ura3Δ0*) was grown to exponential phase ($OD_{600} \sim 0.5$) in YPD medium (1% yeast extract, 2% peptone, 2% glucose) for at least 16 h at 30 °C (naive cells). To change cells from YPD to YPGal (1% yeast extract, 2% peptone, 2% galactose) cells were collected by 2 min centrifugation (3000 g) and washed with prewarmed YPGal. After wash, the cells were collected by 2 min centrifugation at 3000 g and resuspended in prewarmed YPGal for 3 h. Next, cells were shifted to glucose-containing media (YPD) for 3 h, performing a wash with prewarmed YPD as previously described. Finally, galactose-primed cells were washed and exposed to prewarmed YPGal. All yeast samples (2 ml) were collected by centrifugation (30 seconds at 8000 x *g*) and pellets were frozen in liquid nitrogen. *Schizosaccharomyces pombe* (h-) used as spike-in was grown at 30 °C to mid-log phase (OD600 ~0.5)

For library construction, total RNA concentration was measured with Qubit and RNA quality was checked by capillary electrophoresis. We used 2.5 µg total *S. cerevisiae* RNA supplemented with 0.6 ng SIRV-SET3 (Lexogen) as spike-in. rRNA was depleted with illumina Ribo-Zero Gold rRNA Removal Kit (Yeast) according to manufacturer instructions, detailed kit info is provided in Supplementary Data 6. Then, a strand-specific RNA-Seq library was prepared using NEBNext Ultra Directional RNA Library Prep Kit for Illumina following the manufacturer instruction. Briefly, rRNA-depleted RNA was first fragmented and then we used random primer to generate first cDNA strand. dUTP was incorporated into cDNA during the following second-strand synthesis. After end repair and dA tailing, Illumina adaptors were ligated. The second strand containing dUTP was removed using USER enzyme mix. Strand-specific library was prepared with 7 PCR cycles. Library quality was assessed via Qubit and Bioanalyzer. The libraries were sequenced using an Illumina Nextseq 500 instrument.

## Processing, analysis and graphic display of RNA-seq data

*S. cerevisiae* genome assembly and annotation for the RNA-Seq data analysis was downloaded from SGD database (version 64-1-1) and annotation from (Xu et al., 2009). For *S. pombe* we used genome version ASM294v2.20. The quality of the RNA-Seq data was assessed with FastQC (Andrews, 2010). Reads were aligned to the transcriptome by STAR(v2.5.3a) with parameters "--outFilterMismatchNmax 4 --alignIntronMin 13 --alignIntronMax 2482".

Reverse stranded reads were then summarized into gene expression values by featurecounts with parameter "-s 2 -C". Chimeric fragments were excluded from fragment counting.

Read counts were normalized by sum of coding transcriptome. Differential gene expression analysis was perfomed using the DESeq2 (v1.26.0) package in R (v3.6.3). When calculating log fold change of different time points, we use the time 0 of first induction as reference point.

PCA plot data was calculated with plotPCA function of DESeq2 package then plotted with ggplot2. MA plot data was calculated with plotMA function of DESeq2 and then plotted with ggplot2.

Stepwise Annotation for the gene category: first sort gene into 3 groups (Induced genes, genes with no change and repressed genes) then into 5 groups according to memory pattern (induction memory genes, induced genes without memory, genes with no change, repressed genes without memory and genes with repression memory).

Induced genes are defined as genes whose (lfc, $log_2$ fold change) $lfc^{3h} > 0$, $lfc^{1h'} > 0$, $lfc^{3h} > lfc^{1h}$, $lfc^{3h} > lfc^{30min}$ and adjusted $p$ value < 0.001. Induction memory genes are defined as induced genes whose $lfc^{30min'} - lfc^{30min} > log_2(1.5)$ or $lfc^{1h'} - lfc^{1h} > log_2(1.5)$. Repressed genes are defined as genes whose $lfc^{3h} < 0$, $lfc^{1h'} < 0$, $lfc^{3h} < lfc^{1h}$, $lfc^{3h} < lfc^{30min}$ and adjusted $p$ value < 0.001. Repression memory genes are defined as those repressed genes whose $lfc^{30min'} - lfc^{30min} < -log_2(1.5)$ or $lfc^{1h'} - lfc^{1h} < -log_2(1.5)$. $lfc^{1h'}$ and $lfc^{30min'}$ are the $log_2$ fold change of 1h and 30 min in the second induction (primed state) while $lfc^{30min}$, $lfc^{1h}$ and $lfc^{3h}$ are the $log_2$ fold change of 30 min, 1 h and 3 h in the first induction (naive state). For memory genes affected by *RRP6* depletion, we first normalized the gene $log_2$ fold change in primed state by the gene $log_2$ fold change of the longest induction time point (3 h) in the naive state to control for the effect of the mutation and have a fair comparison of memory effect between strains.

Heatmap was generated with Complexheatmap (v2.2.0). $TM_{score}$ was defined and calculated as a measure of relative change amplitude at 15 min in primed state normalized by RNA abundance of 3 h in the naive state. For induced memory genes, the memory index is illustrated as $lfc^{15min'} - lfc^{3h}$. For repression memory genes, the memory index is illustrated as $lfc^{15min'} - lfc^{3h}$.

Bigwig file for IGV visualization was generated with deeptools(v3.1.0) bamcoverage with normalization factor generated by inverting size factor generated in DESeq2 normalized by sum of coding transcriptome. Hypergeometric test was performed using https://systems.crump.ucla.edu/hypergeometric/. GO enrichment was performed with R package "clusterProfiler". mRNA codon stability index, translation efficiency were obtained from Carneiro et al.[45].

## MNase Seq and ChIPseq experiment and library preparation

We analyzed *S. cerevisiae* cells (wildtype and *rrp6Δ*) and used *S. pombe* as spike-in. Cell cultures were grown until $OD_{600}$ 0.3–0.5 (*S. cerevisiae*) or 0.74 (*S. pombe*). 100 ml culture per *S. cerevisiae* sample and 30 ml of S. *pombe* culture was crosslinked using 1% formaldehyde for 15 minutes at room temperature. Formaldehyde was quenched by 0.125 M glycine for 5 minutes. Then cells were washed three times with cold TBS, flash-frozen in liquid nitrogen and stored at −80 °C. Frozen cell pellets were resuspended in zymolyase solution (1 M sorbitol, 50 mM Tris-HCl pH 7.5, 1% beta-mercaptoethanol, 0.1 U/µl zymolyase). The zymolyase digestion proceeded at +37 °C for 30 min (*S. cerevisiae*) or 90 min (*S. pombe*). Spheroplasts were isolated by centrifugation at 6000 x g for 10 minutes at +4 °C. Spheroplasts were resuspended in NP buffer (10 mM Tris-HCl pH 7.5, 1 M sorbitol, 50 mM NaCl, 5 mM MgCl₂, 1 mM CaCl₂, 0.075% NP-40 (Tergitol), 1% beta-mercaptoethanol, 0.5 mM spermidine, 1% yeast protease inhibitor cocktail). The suspensions were pre-warmed and then 0.5 U/µl of MNase for *S. cerevisiae* and 2 U/µl for *S. pombe* was added per sample. MNase digestion proceeded at 37 °C for 40 min (*S. cerevisiae*) or 30 min (*S. pombe*) and stopped by addition of EGTA. The supernatants containing chromatin fragments were collected and diluted with 1 ml RIPA buffer (10 mM Tris-HCl pH 8.0, 1 mM EDTA pH 8.0, 0.1% SDS, 140 mM NaCl, 1% Triton X-100, 0.1%

sodium deoxycholate) with 1% yeast protease inhibitor cocktail. Equal volumes of *S. pombe* chromatin were spiked into *S. cerevisiae* chromatin samples at this point, corresponding to around 3% of *S. cerevisiae* DNA. Each sample was immunoprecipitated with anti-H3K4me3 (Abcam ab8580) and anti-H3K4me2 (Abcam ab7766) antibodies. Protein A/G beads (Pierce) were coupled to antibodies (3 µg/sample) and resuspended in diluted chromatin and rotated o/n at 4 °C. The next day, beads were washed with RIPA buffer, RIPA−500 buffer (10 mM Tris-HCl pH 8.0, 1 mM EDTA pH 8.0, 0.1% SDS, 500 mM NaCl, 1% Triton X-100, 0.1% sodium deoxycholate), LiCl wash buffer (10 mM Tris-HCl pH 8, 1 mM EDTA, 250 mM LiCl, 0.5% v/v NP-40, 0.5% w/v sodium deoxycholate), and TE buffer (10 mM Tris-HCl pH 8, 1 mM EDTA pH 8). Volume of each wash was 150 µl. Chromatin was eluted from beads in $2 \times 10$ µl ChIP elution buffer (Tris-HCl pH 8 50 mM, 1% SDS, 10 mM EDTA pH 8). The immunoprecipitated chromatin as well as a 20 µl sample of input chromatin were decrosslinked by TE, RNase cocktail, Proteinase K and 6 µl SDS (10% w/v), incubating overnight at 65 °C. DNA was purified by ethanol precipitation. Illumina sequencing libraries were prepared from the DNA using the NEBNext Ultra II kit without dual size selection and using 1.4X volume of AMPure XP beads for the purification steps. The libraries were sequenced on Illumina's NextSeq 500, paired-end, 39 bases from each end. Data was collected using NextSeq 500 default software.

## Mnase seq and ChIP analysis

Illumina adaptor sequences were detected and trimmed using Trim-Galore. Trimmed reads were then aligned to *Saccharomyces cerevisiae* genome assembly R64-1-1 using bwa (v0.7.17). Highly repetitive Ribosomal DNA regions were removed from alignment. PCR duplicates were marked and removed by Picard MarkDuplicates. Deduplicated reads from biological replicates were then merged together. Bigwig files were generated from merged bam files by deeptools bamcoverage command with parameters "--binSize 1 --MNase --minFragmentLength 100 --maxFragmentLength 200 --normalizeUsing CPM". These parameters aimed to take only mononucleosome, deconvolute and take only the center dyad genomic coordinate of each nucleosome. Bed files containing gene groups generated from RNAseq data were provided to deeptools computematrix command to extract the nucleosome occupancy and average histone modification level of each gene within particular group. Deeptools plotProfile tool was then used to summarize above matrix into metagene plot with parameter "–perGroup".

## Reanalysis of CRAC data

We used published datasets from Bresson et al. 2017 (PolII, Nab3 and Mtr4), Clémentine Delan-Forino et al. 2020 (Trf4 and Trf5) and Tuck et al. 2013 (Ski2) with GEO accession number: GSE86483[30], GSE135526[37], GSE46742[38]. Provided bigwig files were quantified with multiBigwigSummary tools from deeptools[46].

For CRAC bedgraph files quantification (Mtr4, Nab3 and RNApolII) we followed the following: 1) the bedgraph files were converted to bed files. 2) the score column in the converted bed file was expanded as the corresponding number of reads (to make one read as one row). 3) then bed files were quantified using bedtools coverage to get counts number for each feature. We discarded those genes with less than 20 counts. Next, we normalized gene coverage by the total counts of each sample. Finally, we normalized the intrinsic association of each decay factor by RNApolII counts. To normalize the Trf4, Trf5, and Ski2 CRAC data, we used the PolII CRAC counts which were generated in the same biological settings from Bresson et al. 2017 paper.

For CRAC data in sgr files format (Ski2), we converted the sgr file to plus-strand bed file and minus strand bed file first. Then we expand the score column in bed file to reads records (make one read as one row). Finally, we quantified the reads counts of each feature from the

bed file using bedtools coverage. We discarded those genes with less than 20 counts.

For CRAC data in gtf format (Trf4, Trf5), gtf files were first converted to bed12 files with gtf2bed from ea-utils[47]. Bed12 files were then converted to bed6 files. Bed6 files were then quantified with bedtools coverage. We discarded those genes with less than 20 counts.

## Metabolic labeling and SLAM-Seq

Metabolic labeling of newly synthesized RNA molecules was performed as previously described[40]. Briefly, 4-thiouracil (4tU) (Sigma) was dissolved in NaOH (83 mM). Newly synthesized RNA was labeled for 10 minutes at a final concentration of 5 mM 4-tU. MES buffer (pH 5.9) with a final concentration 10 mM was added to media to avoid pH change as a result of NaOH addition. At each time points before harvesting ($t_0$, $t_{30}$, $t_0'$, $t_{30}'$ for both wild type and *rrp6Δ*; $t_0$, $t_0'$ for both *ski2Δ* and *xrn1Δ*), prewarmed 4tU was added to culture media (YPD with MES buffer or YPGal with MES buffer) 10 min ahead of desired time point. During the last minute of labeling, cells were collected by centrifugation at 3000 g and snap frozen in liquid nitrogen immediately. RNA was extracted and purified with MasterPure Yeast RNA Purification Kit. Total RNA was then subjected to thiol(SH)-linked alkylation by iodoacetamide (final 0.5 M) at 50 °C for 15 minutes[39], the reaction was stopped by adding 0.1 M DTT to the final concentration at 20 mM. RNA was cleaned by ethanol precipitation. rRNA was depleted using the RiboPools Depletion Kit (siTOOLs Biotech), strand specific library was prepared by Ultra™ II Directional RNA Library Prep Kit for Illumina® following manufacture instructions except for the use of NEBNext Strand Specificity Reagent. Sequencing was performed with single end setting, read length 121 bp on Illumina Nextseq 500 sequencer. Data was collected using NextSeq 500 default software.

For Pulse and chase experiment, briefly, 4-thiouracil (4tU) (Sigma) was prepared as explained above. Prewarmed 4tU was added to culture media (YPD with MES buffer) containing wild-type cells. After 1 h labeling with 4tU to fully label the whole transcripts, cells were washed and resuspend in YPGal media and time points were collected at 0, 10 and 30 minutes ($t_0$, $t_{10}$, $t_{30}$, $t_0'$, $t_{10}'$, $t_{30}'$) by centrifugation and snap frozen in liquid nitrogen immediately. Between the first and second exposure to YPGal, cells were cultured for 3 h in YPD. Libraries were prepared and sequenced on Illumina Nextseq 500 sequencer. Data was collected using NextSeq 500 default software.

SLAM-Seq data was analyzed with slamdunk provided by nfcore pipeline (v1.0.0). https://nf-co.re/slamseq. As stranded library was prepared with dUTP method, fastq files were first converted to reverse complementary reads to feed into slamdunk nf core pipeline. Adapter contamination and low-quality region was trimmed using TrimGalore (v0.6.5) (trim length 30 bp). Lifted-over transcript annotation for (Xu et al., 2009) to genome version 64-1-1 was downloaded from SGD database. Annotation was first converted into a bed file and used as input for parameter "-utrbed". At least 2 T > C conversions per read was regarded as a confident call for nascent RNA reads. SNP masking was employed to distinguish Single Nucleotide Polymorphism from converted nucleotides. Degradation rate is calculated as $60 * \ln(2) / t_{1/2}$ ($t_{1/2}$ is half-time of gene).

## Sample Preparation for LC-MS

Yeast cells were quenched by adding pure trichloroacetic acid (Sigma Aldrich) to the yeast cultures to a final concentration of 10% (v/v) and incubating for 10 min on ice. Samples were then centrifuged at 2500 g for 5 min at 4 °C and the supernatant was discarded. The pellet was washed twice with 10 ml cold acetone before being transferred into a new tube. After an additional centrifugation step at 3000 g for 5 min at 4 °C, the acetone was removed and the pellet was further processed for protein extraction.

## Cell Lysis and Protein Extraction

To lyse the cells, cell pellets were first mixed with glass beads (Sigma Aldrich) and 500 µl of lysis buffer containing 8 M urea, 50 mM ammonium bicarbonate and 5 mM EDTA (pH 8). The mixture was then transferred to a FastPrep-24TM 5 G Instrument (MP Biomedicals) where cells were disrupted at 4 °C by 5 rounds of beads-beating at 30 seconds with 120 seconds pause between the runs. Samples were then centrifuged for 10 min at 21'000 x *g* to remove cell debris and the supernatants were transferred into a new tube. The protein concentration was determined using the bicinchoninic acid Protein Assay Kit (Thermo Scientific) following the manufacturer's protocol.

## In solution protein digestion

100 µg of protein extracts were subjected to digestion. Samples were vortexed and sonicated for 5 min. In the first step, dithiothreitol (Sigma Aldrich) was added to a final concentration of 5 mM and incubated for 30 min at 37 °C to reduce the disulfide bridges followed by the alkylation of free cysteine residues with iodoacetamide (Sigma Aldrich) at 40 mM final concentration (30 min at 25 °C in the dark). Samples were pre-digested with lysyl endopeptidase (Wako Chemicals) at an enzyme substrate ratio of 1:100 for 4 h at 37 °C and then diluted 1:5 with freshly prepared 0.1 M ammonium bicarbonate to reduce urea concentration to 1.6 M. Sequencing grade trypsin (Promega) was added at an enzyme substrate ratio of 1:100 and digested at 37 °C for 16 h. The digestion was stopped by adding formic acid (Sigma Aldrich) to a final concentration of 2%. The digested samples were loaded onto SepPak C18 columns (Waters) that were previously primed with 100% methanol, washed with 80% acetonitrile (ACN, Sigma Aldrich), 0.1% FA and equilibrated 3 times with a 1% ACN, 0.1% FA solution. The flow-through was loaded once more onto the columns and the peptides bound to C18 resins were afterwards washed 3 times with a 1% ACN, 0.1% FA solution and eluted twice with 300 µl 50% ACN, 0.1% FA. The elution was dried down in a vacuum centrifuge and peptides were resuspended in a 3% ACN, 0.1% FA solution to a concentration of 1 mg/ml before LC-MS analysis.

## Liquid chromatography–mass spectrometry (LC-MS) analysis

Peptide samples were analyzed in a Data-Independent Acquisition mode (DIA) with an Orbitrap Exploris 480 mass spectrometer (Thermo Fisher Scientific) equipped with a nano-electrospray ion source and a nano-flow LC system (Easy-nLC 1200, Thermo Fisher Scientific). Peptides were separated with a 25 cm fused silica capillary column with inner diameter of 75 µm packed in house with 1.9 µm C18 beads (Dr. Maisch Reprosil-Pur 120). For LC fractionation, buffer A was 3% ACN and 0.1% FA and buffer B was 0.1% FA acid in 90% ACN and the peptides were separated by 2 h non-linear gradient at a flow rate of 250 nl/min with increasing volumes of buffer B mixed into buffer A. The DIA-MS acquisition method consisted of a survey MS1 scan from 350 to 1650 m/z at a resolution of 120,000 followed by the acquisition of DIA isolation windows. A total of 40 variable-width DIA segments were acquired at a resolution of 30,000. The DIA isolation setup included a 0.5 m/z overlap between windows.

## Quantitative proteomics data analysis

DIA-MS measurements were analyzed with Spectronaut 16 (Biognosys AG) using direct searches. In brief, retention time prediction type was set to dynamic iRT (adapted variable iRT extraction width for varying iRT precision during the gradient) and correction factor for window 1. Mass calibration was set to local mass calibration. The false discovery rate (FDR) was set to 1% at both the peptide precursor and protein levels. Digestion enzyme specificity was set to Trypsin/P and specific. Search criteria included carbamidomethylation of cysteine as a fixed modification, as well as oxidation of methionine and acetylation (protein N-terminus) as variable modifications. Up to 2 missed cleavages were allowed. The DIA-MS files were searched against the

*Saccharomyces cerevisiae* UniProt fasta database (updated 2021-04-02). Differentially regulated proteins were determined with an unpaired t-test statistic with Storey method correction.

## Reporting summary

Further information on research design is available in the Nature Portfolio Reporting Summary linked to this article.

## Data availability

The RNA-seq, SLAM-Seq ChIP-seq and MNase-seq data generated in this study have been deposited in the GEO database under accession code GSE201036 and GSE218400. Both the raw data and processed data are available at GEO. The raw and processed data generated in this study are also provided in the Supplementary Information/Source Data file. We used published datasets from Bresson *et al.* 2017 (PolII, Nab3 and Mtr4), Clémentine Delan-Forino *et al.* 2020 (Trf4 and Trf5) and Tuck et al. 2013 (Ski2) with GEO accession number: GSE86483[30], GSE135526[37], GSE46742[38]. The mass spectrometry proteomics data have been deposited to the ProteomeXchange Consortium via the PRIDE partner repository with the dataset identifier PXD036586.

## Code availability

Codes are available at https://github.com/PelechanoLab/2022TranscriptionalMemoryLab.

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

## Acknowledgements

We thank all members of the Pelechano, Kutter and Friedländer laboratories for useful discussions. We thank Adrian Cortés-Sanchón for initial support with the transcriptional memory screening. We thank José Enrique Pérez-Ortín and Bastian Linder for useful comments in the manuscript. Computational analysis was performed on resources provided by the Swedish National Infrastructure for Computing (SNIC) through Uppsala Multidisciplinary Center for Advanced Computational Science (UPPMAX) partially funded by the Swedish Research Council through grant agreement no. 2018-05973. We also acknowledge the use of the HPC Cloud Platform of Shandong University. This study was financially supported by the Swedish Research Council (VR 2016-01842, 2020-01480 and 2021-06112), a Wallenberg Academy Fellowship (KAW 2016.0123 and 2021.0167), the Swedish Foundations' Starting Grant (Ragnar Söderberg Foundation) and Karolinska Institutet (SciLifeLab Fellowship, SFO, SDG and KI funds) to VP. VP also acknowledges the support from Swedish Research Council Research Environment Grant (VR 2019-02335), a Joint China-Sweden mobility grant from STINT (CH2018-7750), a grant to Science for Life Laboratory National COVID-19 Research Program funded by the Knut and Alice Wallenberg Foundation (KAW 2020.0241, V-2020-0699), a grant from Vinnova (2020-03620) and to the EDCTP2 programme supported by the European Union (RIA2020EF-3030, RADIATES). IP receives funding from the Helmholtz Young Investigators program of the Helmholtz Association and from the European Research Council (ERC) under the European Union's Horizon 2020 research and innovation programme (grant agreement ERC-STG No 948544). YZ is funded by a fellowship from the China Scholarship Council.

## Author contributions

V.P. and B.L. conceived the study. P.Z., M.M.T. and G.L performed and analyzed the screen with supervision from V.P. and L.M.S. A.A. contributed to the chromatin analysis. Y.Z. and Y.P.S. contributed to the SLAM-Seq and additional work during the revision process. E.F. performed the proteomic analysis under the supervision of I.P. B.L. performed all other experimental and computational work. B.L. and V.P. drafted the original manuscript. All authors reviewed and edited the manuscript. V.P. supervised the project.

## Funding

## Competing interests

The authors declare no competing interests.
