## [Peer Review File · Nature Communications]

Differential regulation of mRNA stability modulates transcriptional memory and facilitates environmental adaptationREVIEWER COMMENTS

Reviewer #1 (Remarks to the Author):

The publication by Li et al. from the Pelechano lab aims at finding novel pathways involved in the so-called “transcriptional memory”. In other words, gene expression of cells that undergo environmental changes can be primed by an initial stimulus and respond faster when the stimulus is repeated. Transcriptional memory has been addressed by many different labs and most studies pointed out chromatin regulation as key in this process. In this manuscript, the authors performed a genetic screen using the well-described GAL1 gene as reporter, which is induced faster when cells undergo a second exposure to a galactose-containing medium. Among the candidates, they focused on RRP6, coding for an RNA exonuclease that is part of the nuclear exosome involved in nuclear surveillance. Using RNA-seq, they define the classes of genes sensitive to RRP6 deletion regarding transcriptional memory, and identify genes that are subject to either induction memory or repression memory. Based on additional analyses, the authors exclude that non-coding transcription and chromatin organization might explain the differences between naïve and primed states. They propose that differences are more likely due to differential binding of the nuclear exosome to the different classes of genes. Finally, they demonstrate differential mRNA turnover when comparing naïve and primed states.

The publication is clearly written and the different genome-wide approaches well explained. The literature related to this topic is fairly and correctly cited. Some observations, such as the differential RNA turnover in naïve versus primed cells are novel and of high interest for a better understanding of the process of transcriptional memory. However, a clear mechanism is still lacking to fully catch the message of this publication, especially concerning the repression memory undergone by a fraction of genes.

Below are more detailed comments and propositions for additional experiments and analyses to improve the manuscript and strengthen some of the current conclusions.

Major comments:

- 1) Figures 1 and 2 and their associated description are crystal clear. However, the analysis of the effect of non-coding transcription presented in Figure 3 could be improved. In Figure 3A, it may be interesting and useful to present the RNA-seq data as the t_0'/t_0 in WT and *rrp6Δ*. Indeed, the problem in showing the Normalized RNA abundance is that the spreading of the values might hide differences that could be significant. It may therefore be more correct to define the t_0'/t_0 ratio of individual genes and then examine whether differences exist between the classes (as done in Figure 5). Would such an approach change the conclusion? Based on Figure 3A, there is for sure a correlation between Enhanced induction memory and a higher amount of ncRNAs in promoters as compared to the other classes of genes.
- 2) Figures 3B-D: It is indeed interesting to perform MNase-seq (or histone modifications analyses) to check any difference between naïve and primed cells. However, strong statistics should be provided before discarding a contribution of chromatin organization. The authors could analyze the fold-change primed/naïve centered on a specific feature (TSS for example) to define whether this may reveal a trend

when looking at the different classes?

The metagene analysis would benefit from a fine smoothing to get a more “regular” profile (the resolution of such an assay is sufficient to propose a fine smoothing of the signal).

It might also be interesting to show the profiles of the Supp. Figure 3D on the main Figure because it is not clear at this point why to compute all the genes and not the different classes. However, the global metagene plot may be sufficient to illustrate the MNase-seq if accompanied by the statistical analyses proposed above.

3) The use of the data from the Tollervey lab in Figure 4 is a good idea. Indeed, it shows that genes with Enhanced induction memory tend to show a higher recruitment of the TRAMP and NNS complexes at steady-state. However, if not mistaken, these data were collected in glucose-containing medium. So far, data of primed cells are not available, and it is not known whether TRAMP and NSS binding changes in galactose. Nevertheless, the abstract seems to claim that these data are in the publication: “show that changes in nuclear surveillance factor association can enhance both gene induction and repression in primed cells”. Thus, some sentences such as “Differential association...modulates transcriptional memory” or “differential binding...contribute to transcription memory” appear as too strong given the fact that they are only based on correlations.

Would it be possible to test the binding of the TRAMP complex in primed cells? One might expect a decrease of the binding in primed cells for genes presenting an induced memory and it would be very nice to show that.

Moreover, since the conclusions above are only based on steady-state values, it is unclear why the increase of ncRNA expression in promoters of enhanced memory induced genes in *rrp6Δ* might not be considered to explain the difference between naïve and primed states. Why discard the steady-state information in one case (Figure 3) and consider it as valid in another case (Figure 4)?

4) The observation of different RNA turnovers in Figure 5 is of high interest. It matches quite well with Figure 4 (although it includes a huge part of cytoplasmic decay). However, in *rrp6Δ*, there is still an overall decrease of RNA turnover in primed cells indicating that something else is involved in the process (probably the cytoplasmic decay as pointed out by the authors). Shouldn't it be a major hypothesis to follow? What makes the turnover different in primed cells? A decrease or a less functional cytoplasmic decay machinery?

This missing point should be addressed to make this manuscript acceptable for Nature Communications.

Minor comments:

Page 12: (reviewed in 12) but 12 is not a review. Double check if some references were not mixed.

Figure 4: It would be nice to acknowledge on the figure that data were taken from the Tollervey lab publication.

Reviewer #2 (Remarks to the Author):

The transcriptional response employed by cells can be affected by previous experiences. This has been most studied in the case of faster/stronger transcriptional induction of genes in response to a second exposure to a stimulus, but it is also true of repression; genes that are repressed by a stimulus can be more rapidly repressed upon a second exposure. Such phenomena are called transcriptional memory (either induction memory or repression memory). The authors set out to identify factors that affect transcriptional induction memory in budding yeast, induced by growth in galactose. They generated a fluorescent reporter for the GAL1 promoter and, using pooled, bar-coded deletion strains, identified mutations that either diminished or enhanced memory. Among the mutants that exhibited enhanced memory, they identified several factors involved in nuclear RNA degradation (Rrp6, Lrp1 and Pap2). RNAseq in *rrp6Δ* strains revealed that this mutation led to enhanced activation of a subset of the genes that exhibit induction memory (88 of 546) as well as enhanced repression of a subset of the genes that exhibit repression memory (158 of 773). This suggests that nuclear RNA degradation constrains both induction memory and repression memory. The authors do not observe a strong connection between galactose memory and nucleosome positioning, histone H3K4 methylation or the accumulation of non-coding RNAs in the *rrp6Δ* mutant. However, using CRAC, they do find a correlation between binding of RNA degradation complexes TRAMP and NNS and genes that exhibit Rrp6-constrained memory: TRAMP and NNS bind most strongly to induced mRNAs that exhibit Rrp6-constrained induction memory and least strongly to mRNAs that exhibit Rrp6-constrained repression memory. Furthermore, the global mRNA turnover rate is reduced in primed cells and decreases the most for the genes that exhibit induction memory and decreases the least for the genes that exhibit repression memory. From these results, the authors conclude that nuclear RNA degradation plays an important role in constraining both the accumulation and decrease in mRNAs during memory.

The data in the paper are generally of high quality and the interpretations mostly reasonable. However, there are several significant questions that are raised but not answered in the manuscript. First, what is the molecular basis for the effects of Rrp6 in constraining induction or repression of subsets of genes? At present, the manuscript stops short of revealing the molecular explanation for this specificity. Second, since Rrp6 is not required for the global decrease in mRNA turnover associated with memory, how does this relate to the role of Rrp6 in memory? This effect is clear and strong, but not dependent on Rrp6. Likewise, the massive global decrease in mRNA turnover upon switching from glucose to galactose seems likely to be relevant, but is unaddressed. Third, given the well-studied role of Gal1 protein in promoting galactose induction memory and fitness and the fact that even very low levels of transcription produce this effect, it is surprising that the authors did not test the hypothesis that *rrp6Δ* leads to slightly more Gal1 protein during the first induction, leading to stronger induction memory. While it may not be possible to address all of these questions experimentally, they should at least be discussed more explicitly. Below, I highlight other points that would strengthen the paper.

Major points

1) Authors conclude that changes in H3K4me2/3 do not correlate with the observed transcriptional memory. However, the metagene plots (Fig. 3 C & D) used to draw this conclusion are for all genes. Authors do provide metagene plots for gene subsets (Fig. S3D) and conclude no obvious changes (qualitative conclusion?). However, these data raise two questions. First, the data in Figure S3D suggests that the average level of H3K4 methylation over inducible genes (i.e. that should be unexpressed) is very similar to that over repressible genes (i.e. that should be expressed). This is a surprising observation and raised concerns about how this sub setting was performed. The authors should provide plots for H3K4me2 and H3K4me3 signal for representative genes with established transcriptional status in glucose to confirm that transcribed genes exhibit greater H3K4 methylation than repressed genes. Also, because H3K4me2/3 ChIP-seq signal may vary across these subsets, a comparison of signal for individual genes between conditions is required to evaluate whether significant changes occurred over some, but not all. Are there any genes for which there is a correlation between H3K4 methylation or nucleosome positions and either induction memory or repression memory?

2) Authors indicate that changes detected by SLAM-seq reflect changes in cytoplasmic mRNA decay (page 10, lines 21-23). mRNA Turnover is calculated as Nascent Reads/Total Reads, and authors conclude that changes in mRNA Turnover between conditions is an indication of increased cytoplasmic stability or targeted degradation. However, SLAM-seq may reflect both changes in mRNA turnover and increases/decreases in transcription and it is unclear why the authors ignore this component. From the data in the paper, direct comparison of CPM-normalized Nascent reads for each gene in naïve (t0) vs. primed (t0') should reveal whether changes in transcription contribute to the observed memory phenomena. The author's ability to infer changes in mRNA stability from SLAM-seq data is dependent on there being no change in transcription. A better way to establish a change in mRNA turnover rates is to measure them directly, using pulse labeling of 4-tU and measuring the rate of replacement of existing mRNAs with 4tU-labeled RNAs (PMC6152797) .

3) Fig 4 suggests that Mtr4 and Nab3 binding was detected to the vast majority of expressed mRNA in your growth conditions: 791/882 induced, 996/1067 repressed, 3200/? unchanged mRNAs. This is surprising as the canonical binding of these proteins is to a subset of mRNAs, CUTs, snRNAs and snoRNAs. This raises concerns about the background of this method and suggests that additional controls are necessary to establish the biological significance of these interactions. Are there sequence motifs in the RNAs that are recovered compared to an input or tagless control? Are ncRNAs/CUTs enriched in these CRAC experiments?

Clarifications and minor concerns

1) Clarify how TMscore is calculated and include it in the main text as well as the figure legend where it is first introduced. Clarity is imperative because the TMscore, which is itself a ratio, is later used again as a ratio of ratios (for instance, see 2 below).

2) The same expression criteria are used to described different sets of genes (page 7, lines 5-8). (TMscore rrp6Δ / TMscore wild-type < 0.667) is used to described 7 genes with attenuated activation memory as well as 158 genes with enhanced repression memory. While I believe induced and repressed categories were subset based on these criteria, these details should be explicitly stated for the reader.

3) In the methods, please provide relevant details for CRAC and subsequent analysis, even if identical to Reference 29. At a minimum, provide: number of cells crosslinked, crosslinking-time, antibodies used,

amount of RNA used for library prep, library prep kit used, sequencing platform and read-length. In addition, provide details for how the analysis was performed for data illustrated in Figure 4. For instance: how many reads were required for a given gene to be included in the plot (minimum threshold)?

4) Include the wild type strain and a legend in Fig 1C.

5) For Figure 1A, consider using a different color scheme. The one used in the original manuscript is going to be difficult to see for red-green colorblind readers.

6) For Figure 2A, two suggestions: first, the colors are not intuitive to represent expression with a dark color and repression with a light color. Second, it would be very helpful to add a plot of the mean expression (as in panel C) for all 6 categories of genes.

7) 540 genes were differentially expressed in wild type vs *rrp6Δ* in naïve conditions. Please categorize these genes with GOterm enrichment and conclude how much, if any, of the memory genes seen are a result of the differential expression of the strains themselves.

8) When calculating TMscore, explain why an absolute value of 5 was added or subtracted from induced and repressed memory genes, respectively (page 21, lines 6 &7)? The plotted values of TMscore (Fig 2D) do not suggest ± 5 .

Reviewer #3 (Remarks to the Author):

In this very interesting manuscript, the authors show that in addition to transcriptional memory happening at chromatin level that have been previously described, post-transcriptional memory contributes to the adaptation of *S cerevisiae* to a changing environment. They developed a genome wide screening system based on a reporter using GFP under the control of GAL1 promoter and transformed it in the deletion library : it appears as an ingenious and robust tool to highlight factors involved in galactose adaptation memory. After this screening, they pointed out some factors from the TRAMP and nuclear exosome complex, as enhancing memory. Then, they generated data from global methods including RNA seq and SLAM-seq, and used published RNA interactome obtained by CRAC for the factors of interest to understand the contribution of RNA processing to post-transcriptional memory. I believe the results presented here bring new conclusions about adaptation memory, will be of general interest and should be published. However a few points would need to be discussed/addressed before publication :

In the last part of their results, the authors suggest that changes in cytoplasmic mRNA stability contribute to the transcriptional memory phenotypes. Even if the SLAM seq data supports this hypothesis, I am not sure there is enough evidence to affirm this. They could probably use available data to strengthen their argument:

-Could they discuss how cytoplasmic rRNA processing factors were behaving in the reporter assay? We would expect to find Xrn1 and the Ski complex in the memory enhancing factors. Do they find the other subunits of TRAMP and exosome?

-CRAC data in naive conditions are available for the SKI complex (Tuck et al, 2013). Could they do a

similar analysis than they did with Mtr4? Same for other processing factors as Trf4 and Trf5 (CRAC is available too : Delan-Forino et al, 2020). It would be interesting to check the dependance of these transcripts on Pap2 and Trf5 binding. Pap2 and Trf5 behaviour being different on mRNAs, it would be a good control to assess specificity of Pap2.

-The published work (Delan-Forino et al 2017) discriminate mRNA subjected to nuclear degradation and the ones subjected to cytoplasmic degradation. Could the authors look at how the top mRNAs sensitive to RNA processing behave regarding post-transcriptional memory to assess contribution of both cytoplasmic and nuclear memory.

Clarification needed:

-Fig 1C is unclear to me. Are red and black lines 2 independent replicates? Is Y axis representing the ratio with WT expression? If yes it should be stated.

-Is FigS2F about genes with induction memory in WT or delta Rrp6 because it is called for both in the text.

-Fig S2D,E,F,G : Are all the GO terms found for each category of genes (repressed, induced, ...) represented in the figure or just the Top 16 or 17 GO terms are shown? not clear.

Minor changes:-Fig 2C is not cited in the text-p3 l25 little m in mCherry-p14 l13 Majuscule I in "In" to remove-when several papers are cited together, it happens several times in the manuscripts that the numbers are not separated by comma.

Point by point response

Differential regulation of mRNA stability modulates transcriptional memory and facilitates environmental adaptation.

Li et al. 2022

Reviewer #1 (Remarks to the Author):

The publication by Li et al. from the Pelechano lab aims at finding novel pathways involved in the so-called “transcriptional memory”. In other words, gene expression of cells that undergo environmental changes can be primed by an initial stimulus and respond faster when the stimulus is repeated. Transcriptional memory has been addressed by many different labs and most studies pointed out chromatin regulation as key in this process. In this manuscript, the authors performed a genetic screen using the well-described GAL1 gene as reporter, which is induced faster when cells undergo a second exposure to a galactose-containing medium. Among the candidates, they focused on RRP6, coding for an RNA exonuclease that is part of the nuclear exosome involved in nuclear surveillance. Using RNA-seq, they define the classes of genes sensitive to RRP6 deletion regarding transcriptional memory, and identify genes that are subject to either induction memory or repression memory. Based on additional analyses, the authors exclude that non-coding transcription and chromatin organization might explain the differences between naïve and primed states. They propose that differences are more likely due to differential binding of the nuclear exosome to the different classes of genes. Finally, they demonstrate differential mRNA turnover when comparing naïve and primed states. The publication is clearly written and the different genome-wide approaches well explained. The literature related to this topic is fairly and correctly cited. Some observations, such as the differential RNA turnover in naïve versus primed cells are novel and of high interest for a better understanding of the process of transcriptional memory. However, a clear mechanism is still lacking to fully catch the message of this publication, especially concerning the repression memory undergone by a fraction of genes.

We thank the reviewer for his/her positive comments. To strengthen the mechanistic insight of our work we have expanded the work in multiple ways.

- We have performed a proteomic analysis of wild-type and *rrp6Δ* cells in naïve and primed conditions.
- We have performed additional SLAM-Seq experiments. We investigate changes in mRNA stability in *rrp6Δ*, *xrn1Δ* and *ski2Δ*. We have also expanded our analysis of mRNA decay in the wild-type strain.
- We have increased the analysis regarding chromatin, non-coding transcription and CRAC as suggested by all the reviewers. Please see below for details.

Below are more detailed comments and propositions for additional experiments and analyses to improve the manuscript and strengthen some of the current conclusions.

Major comments:

1) Figures 1 and 2 and their associated description are crystal clear. However, the analysis of the effect of non-coding transcription presented in Figure 3 could be improved. In Figure 3A, it may be interesting and useful to present the RNA-seq data as the t_0'/t_0 in WT and *rrp6Δ*. Indeed, the problem in showing the Normalized RNA abundance is that the spreading of the values might hide differences that could be significant. It may therefore be more correct to define the t_0'/t_0 ratio of individual genes and then examine whether differences exist between the classes (as done in Figure 5). Would such an approach change the conclusion? Based on Figure 3A, there is for sure a correlation between

Enhanced induction memory and a higher amount of ncRNAs in promoters as compared to the other classes of genes.

We performed the suggested analysis, shown now in Supplementary Figure S3E-F. As the reviewer pointed, by defining the ratio t_0'/t_0 we can now control for gene-specific features and focus on the naïve to prime differences. We observed subtle differences regarding promoter-proximal accumulation of ncRNA. Although those results do not change the main conclusion of our manuscript, but certainly they add additional detail that we think should be discussed. To make those points clear to the reader, we have modified the text in the manuscript as follows (page 8):

*“To better study the difference between gene groups we defined the ratio t_0'/t_0 for each gene and compared the relative differences for each group (Fig. S3E). Genes with repression memory had slightly lower non-coding transcription over the promoters in both the wild-type and *rrp6Δ* strain in comparison to repressed genes without memory. When investigating antisense transcripts covering promoter regions, we observed a general increase in RNA-Seq coverage for most groups in the *rrp6Δ* strain (Fig. 3A), as expected from antisense CUT accumulation^{24,25}. Next, we investigated the differences in antisense transcription comparing the ratio t_0'/t_0 for each gene. We observed that genes with induction memory enhanced after RRP6 depletion displayed an increased promoter antisense transcription in the wild-type strain, while that was not the case in the *rrp6Δ* strain (Fig. S3F). As we observed only subtle differences between naïve and primed conditions, we concluded that promoter-proximal ncRNA accumulation was likely not the main driver in the formation of transcriptional memory.”*

2) Figures 3B-D: It is indeed interesting to perform MNase-seq (or histone modifications analyses) to check any difference between naïve and primed cells. However, strong statistics should be provided before discarding a contribution of chromatin organization. The authors could analyze the fold-change primed/naïve centered on a specific feature (TSS for example) to define whether this may reveal a trend when looking at the different classes?

As suggested by the reviewer, we analysed the fold-change between primed and naïve conditions centring on specific features. We defined two regions of interest, one covering the +1 nucleosome (0 to 150bp from TSS) and another covering the Nucleosome Depleted region (NDR, -150 to 0 bp from TSS). We provide the analysis of these new data in a new Supplementary Figure (Fig S5).

Regarding MNase coverage, we observe an increase of the +1 nucleosome occupancy in the primed wild-type strain for all gene groups. However, that difference was not clear in the *rrp6Δ* strain (which has already a higher +1 relative occupancy in both naïve and primed states). This change in the wild-type strain affects all gene groups (*i.e.*, with/without induction/repression memory). As we do not observe significant differences when comparing the ratio t_0'/t_0 across groups with different memory behaviour, we conclude that the observed differences +1 nucleosome abundance do not explain the observed phenotypes. Next, we investigated potential changes in the NDR region. As expected, the MNase coverage there was much lower. Interestingly, we observed that genes with repression memory enhanced by RRP6 depletion presented a clear decrease of NDR occupancy in primed conditions in the wild-type strain, but not in *rrp6Δ* strain. To reflect that information, we added the following text (page 9):

*To investigate subtle difference between gene groups we compared MNase coverage for t_0' and t_0 for the +1 nucleosome (0 to 150bp from TSS) and for the Nucleosome Depleted region (NDR, -150 to 0 bp from TSS). The coverage for a +1 nucleosome increased for all analysed genes group in primed conditions for the wild-type strain. For the *rrp6Δ* strain, MNase coverage was in general higher but without significant differences between naïve and primed conditions (Fig S5A). As expected, MNase coverage in the NDR was lower across all regions (Fig S5B). To better study the difference between gene groups we next defined the ratio t_0'/t_0*

*for each gene. We did not observe clear differences for t_0'/t_0 ratio for the +1 nucleosome across groups (Fig S5A). However, genes with repression memory enhanced by RRP6 depletion presented a clear decrease of NDR occupancy in primed conditions in the wild-type strain, but not in *rrp6Δ* strain.*

Next, we performed a similar analysis for H3K4me2/me3. We observed some subtle differences across groups, however the described chromatin results alone were not as clear as the ones we describe for changes in RNA degradation. Independent of that, to make clear the potential contribution of chromatin changes to this phenomenon, we added the following text (page 9).

*“Although in some cases we identified some subtle differences between groups of genes and between strains (Fig. S4A), we did not identify clear differences for +1 nucleosome or NDR coverage between naïve and primed states (Fig S5C-F). However, when investigating difference between gene groups using gene-specific coverage ratio t_0'/t_0 we observed that genes with induction memory present relatively lower H3K4me2 in primed conditions in the *rrp6Δ* strain. In summary, although we observed subtle chromatin differences across groups, those differences alone did not clearly explain the differential transcriptional memory observed in the *rrp6Δ* strain.”*

The metagene analysis would benefit from a fine smoothing to get a more “regular” profile (the resolution of such an assay is sufficient to propose a fine smoothing of the signal).

As suggested by the reviewer, we have implemented a finer smoothing for Fig 3B-D. We used a 5 bp sliding window and plotted the signal every nucleotide.

It might also be interesting to show the profiles of the Supp. Figure 3D on the main Figure because it is not clear at this point why to compute all the genes and not the different classes. However, the global metagene plot may be sufficient to illustrate the MNase-seq if accompanied by the statistical analyses proposed above.

As the analysis that the reviewer suggested have generated significant information, we have decided to split the previous Supplementary figure S3 in two parts: New Supplementary Figure S3 (focusing on the RNA part) and S4 (chromatin). Additionally, we added a new figure S5 to show the detailed chromatin differences between gene groups.

As in this revised version of the manuscript we have also significantly increased our work regarding mRNA decay, we would prefer to keep the additional chromatin information in the supplementary material. We think that that will facilitate the reading of the manuscript.

3) The use of the data from the Tollervey lab in Figure 4 is a good idea. Indeed, it shows that genes with Enhanced induction memory tend to show a higher recruitment of the TRAMP and NNS complexes at steady-state. However, if not mistaken, these data were collected in glucose-containing medium. So far, data of primed cells are not available, and it is not known whether TRAMP and NSS binding changes in galactose. Nevertheless, the abstract seems to claim that these data are in the publication: “show that changes in nuclear surveillance factor association can enhance both gene induction and repression in primed cells”. Thus, some sentences such as “Differential association...modulates transcriptional memory” or “differential binding...contribute to transcription memory” appear as too strong given the fact that they are only based on correlations. Would it be possible to test the binding of the TRAMP complex in primed cells? One might expect a decrease of the binding in primed cells for genes presenting an induced memory and it would be very nice to show that.

Moreover, since the conclusions above are only based on steady-state values, it is unclear why the increase of ncRNA expression in promoters of enhanced memory induced genes in *rrp6Δ* might not be considered to explain the difference between naïve and primed states. Why discard the steady-state information in one case (Figure 3) and consider it as valid in another case (Figure 4)?

We agree with the reviewer that that experiment would be very interesting. However, as we expect to see only subtle changes between naïve and primed conditions, we fear such experiment will not provide data of sufficient quality to either prove or disprove our hypothesis. CRAC experiments (and RNA Binding protein (RBP) immunoprecipitation methods in general) have a relatively low signal-to-noise ratio, even in the hands of the labs which developed them. Those approaches are ideal to identify RBP-mRNA pairs, however the measure of subtle differences between 2 conditions (as we expect between naïve and primed conditions) would be extremely challenging.

However, to address the reviewer's concern, and increase our mechanistic understanding of mRNA degradation in naïve and primed conditions we have investigated the changes in protein abundance. Our working model suggested differential association between RBP and mRNA targets could explain the observed differences in transcriptional memory. However, those differences could be caused by two alternatives (and not exclusive mechanism): 1) by changes in intrinsic association between the mRNA-RBP, or 2) by changes in the abundance of the RBP. In fact, the second options (changes in RBP abundance) is a simpler way to explain our observations. As changes in RBP abundance cannot be directly studied by CRAC, we have performed a MS based quantification of the proteome for naïve and primed cells (Fig 4U-V). This new analysis shows a clear decrease of the relative abundance of component of the TRAMP, NNS and nuclear exosome components in primed conditions. This observation is consistent with our working model where those genes more sensitive to the action of nuclear decay (*i.e.*, genes displaying induction memory) will be relatively stabilized in comparison with the rest of the transcriptome.

Interestingly, and consistent with our SLAM-Seq observation of the first version of the manuscript, we also observed a decrease of factors involved in cytoplasmic decay. However, the decrease of cytoplasmic factors was more modest than the observed for the nuclear decay components.

Independently of our new results, we changed the phrasing of the sentences highlighted by the reviewer to make clear that we are not measuring changes in mRNA-RBP binding between conditions (just intrinsic affinity and changes in RNA degradation machinery in the naïve state). We hope that the additional proteomic data obtained in both naïve and primed conditions, combined with the steady-state information for CRAC in naïve conditions, provides a stronger support for our working model. We summarized all the new proteomic information in Fig 4U-V, Supplementary Figure S6, Supplementary Dataset 4 and the following text:

*“The differential association between an RNA binding protein (RBP) and their mRNA targets can be caused by two alternatives (and not exclusive) mechanism: 1) by changes in intrinsic mRNA-RBP affinity, or 2) by changes in RBP abundance. To test this second possibility, we performed a proteomic analysis of wild-type cells in naïve and primed conditions (Fig 4U and Supplementary Data 4). This analysis revealed a clear decrease of the components of the TRAMP, NNS and nuclear exosome components in primed conditions. The decrease of components involved in nuclear decay would be consistent with a relative increase in expression for those genes more sensitive to its action (e.g., genes displaying induction memory). Unexpectedly, we also observed a relative decrease of factors involved in cytoplasmic decay, although to a lower degree than the nuclear ones (Fig. 4V). However, that was not so clear in the *rrp6Δ* strain (Fig. S6A-B). This suggests that changes in the cytoplasmic mRNA decay may also contribute to modulate transcriptional memory.”*

4) The observation of different RNA turnovers in Figure 5 is of high interest. It matches quite well with Figure 4 (although it includes a huge part of cytoplasmic decay). However, in *rrp6Δ*, there is still an overall decrease of RNA turnover in primed cells indicating that something else is involved in the process (probably the cytoplasmic decay as pointed out by the authors). Shouldn't it be a major hypothesis to follow? What makes the turnover different in primed cells? A decrease or a less functional cytoplasmic decay machinery?

This missing point should be addressed to make this manuscript acceptable for Nature Communications.

We agree that investigating what makes RNA turnover different in naïve and primed cells is the most promising line of research derived from our manuscript. To provide a more complete understanding of this process we have performed two major experiments in this revised version. We have: (i) investigated the changes in abundance for the mRNA decay machinery using proteomics (as discussed before) and (ii) performed a dissection of changes in mRNA turnover with additional mutants involved in cytoplasmic mRNA decay

As the reviewer suspected, there is a decrease in cytoplasmic decay machinery in primed conditions. With the additional data added in this revised version, we can observe a decrease in relative XRN1 protein abundance (Fig 4V). As we have already described the bulk of our proteomic analysis in the previous reply we will not expand here (please see above). In addition, to investigate how the proteome of naïve and primed cells differed, we have also investigated the functional consequences of those changes. Specifically, we have investigated how mRNA stability differs between naïve and primed conditions for mutants of both nuclear and cytoplasmic decay (*xrn1Δ* and a *ski2Δ*). This experiment was performed also in response to the comments raised by reviewers #2 and #3.

In agreement with our proteomic data, we can see that the differential activity of XRN1 is essential to generate an altered turnover in primed cells. Applying SLAM-Seq to the *xrn1Δ* strain, we can see that mRNA stability does not increase in primed conditions. Interestingly, we see that despite the lack of general stabilization, the bulk of the group-specific changes in mRNA turnover remain. This suggests, that even if XRN1 is the main driver of global changes in stability, the action of other mRNA degradation pathways (e.g., nuclear decay) fine-tunes the mRNA-specific stability of the groups of genes with different transcriptional memory. We have added all this information in an expanded Figure 5 and a new Fig S8. We also have expanded with the following text:

*“To further dissect the contribution of cytoplasmic factors to this process, we investigate global changes in mRNA turnover in *ski2Δ* (component of the cytoplasmic 3'-5' exosome) and *xrn1Δ* (cytoplasmic 5'-3' exonuclease) strains. When investigating the *ski2Δ* strain, we also observed a general stabilization of mRNAs in primed cell (Fig. 5I). Like in the wild-type strain, genes displaying induction memory increased in their relative mRNA stability (slower turnover) in primed cells, while genes with repression memory decreased their relative RNA stability (faster turnover) in primed cells (Fig. 5J-L and S8D-E). Finally, we investigated mRNA turnover in *xrn1Δ* cells. As expected, deletion of XRN1 led to a massive mRNA stabilization (Fig. 5M), but contrary to in the previous cases, mRNA in primed cells was less stable than in naïve conditions ($p < 2.2 \cdot 10^{-16}$). However, despite this lack of stabilization of mRNA in primed conditions, when comparing gene-specific differences between naïve and primed conditions we observed the same group-specific relative changes in mRNA turnover as for the other strains (i.e., induction memory genes increasing their relative mRNA stability in primed cells and repression memory decreasing it) (Fig. 5N-P and S8F-G). This shows that cytoplasmic 5'-3' mRNA decay is not essential for most gene specific effects, and rather has a role on global stability change between naïve and primed states. However, and in contrast to what we observed in the other strains, genes with repression memory enhanced by RRP6 depletion presented a relative decrease in their mRNA stability in primed cells compared to other repression memory genes (Fig. 5P, $p < 1.123 \cdot 10^{-7}$). Interestingly those same genes present a particularly high codon adaptation index (Fig. S8H). As, codon optimality has been*

shown to be a main player controlling cytoplasmic mRNA stability⁴², our results suggest that those genes are more dependent on the action of XRNI, and potentially codon optimality, to regulate their stability between naïve and primed conditions. Taken together, our results show that changes in mRNA stability between naïve and primed states contribute to the transcriptional memory phenotype. And that both nuclear and cytoplasmic mRNA degradation contribute to this process.”

And extended discussion:

*To further dissect the role of other degradation pathways we measured mRNA turnover in a deletion strain for a component of the cytoplasmic 3'-5' exosome (*ski2Δ*) and for the major cytoplasmic 5'-3' exonuclease (*xrn1Δ*). In the case of the *ski2Δ* strain we observed similar results as in the *rrp6Δ* strain. However, in the *xrn1Δ* strain the general stabilization of mRNAs in primed conditions was lost (Fig. 5M). This suggests that the differential activity for XRNI, considered to be the main degradation pathway, is essential to achieve the global differences in mRNA stability between naïve and prime conditions. Reassuringly, the general changes observed in mRNA turnover in the *xrn1Δ* strain are consistent with the relative decrease of Xrn1p abundance that we observed in primed conditions in the wild-type strain. However, we did not capture XRNI in our initial screen as that strain was below our detection limit (Supplementary Dataset S1). Despite the clear role of XRNI controlling global mRNA stability, the relative changes in mRNA turnover for groups with differential transcription memory between naïve and primed conditions was largely maintained (Fig. 5N-P). Interestingly those genes with higher relative destabilization in primed conditions present a particularly high codon adaptation index (Fig. S8H) and is differentially regulated in the *xrn1Δ* strain (Fig. 5P). This suggests that they could be more sensitive to drastic changes in translation (and thus co-translationally regulated cytoplasmic mRNA stability). However, more research would be required to investigate this possibility.*

Minor comments:

Page 12: (reviewed in 12) but 12 is not a review. Double check if some references were not mixed.

We have changed the text and revised the citation (D'Urso *et al.* 2017).

Figure 4: It would be nice to acknowledge on the figure that data were taken from the Tollervey lab publication.

We have expanded the citation in the figure legend with text and not only with a reference number to make that point clear to the reader.

Reviewer #2 (Remarks to the Author):

The transcriptional response employed by cells can be affected by previous experiences. This has been most studied in the case of faster/stronger transcriptional induction of genes in response to a second exposure to a stimulus, but it is also true of repression; genes that are repressed by a stimulus can be more rapidly repressed upon a second exposure. Such phenomena are called transcriptional memory (either induction memory or repression memory). The authors set out to identify factors that affect transcriptional induction memory in budding yeast, induced by growth in galactose. They

generated a fluorescent reporter for the GAL1 promoter and, using pooled, bar-coded deletion strains, identified mutations that either diminished or enhanced memory. Among the mutants that exhibited enhanced memory, they identified several factors involved in nuclear RNA degradation (Rrp6, Lrp1 and Pap2). RNAseq in *rrp6Δ* strains revealed that this mutation led to enhanced activation of a subset of the genes that exhibit induction memory (88 of 546) as well as enhanced repression of a subset of the genes that exhibit repression memory (158 of 773). This suggests that nuclear RNA degradation constrains both induction memory and repression memory. The authors do not observe a strong connection between galactose memory and nucleosome positioning, histone H3K4 methylation or the accumulation of non-coding RNAs in the *rrp6Δ* mutant. However, using CRAC, they do find a correlation between binding of RNA degradation complexes TRAMP and NNS and genes that exhibit Rrp6-constrained memory: TRAMP and NNS bind most strongly to induced mRNAs that exhibit Rrp6-constrained induction memory and least strongly to mRNAs that exhibit Rrp6-constrained repression memory. Furthermore, the global mRNA turnover rate is reduced in primed cells and decreases the most for the genes that exhibit induction memory and decreases the least for the genes that exhibit repression memory. From these results, the authors conclude that nuclear RNA degradation plays an important role in constraining both the accumulation and decrease in mRNAs during memory.

The data in the paper are generally of high quality and the interpretations mostly reasonable. However, there are several significant questions that are raised but not answered in the manuscript. First, what is the molecular basis for the effects of Rrp6 in constraining induction or repression of subsets of genes? At present, the manuscript stops short of revealing the molecular explanation for this specificity. Second, since Rrp6 is not required for the global decrease in mRNA turnover associated with memory, how does this relate to the role of Rrp6 in memory? This effect is clear and strong, but not dependent on Rrp6. Likewise, the massive global decrease in mRNA turnover upon switching from glucose to galactose seems likely to be relevant, but is unaddressed. Third, given the well-studied role of Gal1 protein in promoting galactose induction memory and fitness and the fact that even very low levels of transcription produce this effect, it is surprising that the authors did not test the hypothesis that *rrp6Δ* leads to slightly more Gal1 protein during the first induction, leading to stronger induction memory. While it may not be possible to address all of these questions experimentally, they should at least be discussed more explicitly. Below, I highlight other points that would strengthen the paper.

We thank the reviewer's careful analysis our work. Although some of the specific points are difficult to address experimentally, as the reviewer already indicates, we tried to improve our analysis of those in this revised manuscript.

1. To improve our molecular understanding of the effect of the RRP6, we have now performed a proteomic analysis of naïve and primed cells (Fig 4U-V and Supplementary Fig S6). This analysis shows a clear decrease of the relative abundance for component involved in nuclear decay (*e.g.* TRAMP, NNS and RRP6) in primed conditions. In this revised manuscript, we describe the existence of an intrinsic differential gene-specific affinity (as revealed by CRAC) combined with the differences in protein abundance for the RNA decay machinery between naïve and primed cells (as revealed by our new proteomic experiments). Those results are consistent with a working model where those genes more sensitive to the action of nuclear decay (*i.e.*, genes displaying induction memory) would be relatively stabilized in comparison with the rest of the transcriptome. Please see response to reviewer #1.3 for extended discussion. In addition to that, we have also performed new SLAM-Seq analysis for different mutants involved in cytoplasmic mRNA decay (please see response to reviewer #1.4) and reanalysis of available CRAC datasets as suggested by reviewer #3 (please see below).

2. Regarding the change in global mRNA stability between primed and naïve cells, in our original submission we suggested that that was likely mainly driven by the effect of XRN1. In this improved version of the manuscript, after performing additional SLAM-Seq experiments, we can confirm that that is the case (new Fig. 5). Our new data show that both the nuclear and cytoplasmic decay

collaborate to generate the observed gene-specific effects. And that, even when deleting nuclear or cytoplasmic mRNA degradation factors, gene specific differences between naïve and primed conditions are maintained. Please see our response to reviewer #1.4 and the changed text in the manuscript (in red).

3. Regarding the apparent massive global decrease in mRNA turnover upon switching from glucose to galactose it is almost certainly caused by the well know massive decrease in transcription upon switch to galactose. To make that point clear we expanded the text and cite (Garcia-Martinez et al Mol Cell 2004, PMID: 15260981). In addition, to better measure the changes in those conditions, and to disentangle the contribution of transcription and mRNA decay (as the reviewer suggest in the major point 2), we have now performed SLAM-Seq analysis using a pulse-chase strategy. Please see below for additional detail.

4. Finally, we took advantage of our newly generated proteomics data in naïve and primed conditions to investigate GAL1. We did not observe clear differences in Gal1 protein accumulation in primed cells between wild-type and *rrp6Δ* strains (new Supplementary Figure S6C). Thus, we discarded the role of cytoplasmic Gal1P accumulation as the mechanism driving the studied differential transcriptional memory phenotype in this condition. We added the following text and a new Figure S6C:

*“Finally, we took advantage of the generated proteomic data to investigate potential role of Gal1p cytoplasmic accumulation in transcriptional memory. It has been shown that cytoplasmic accumulation of Gal1p can facilitate faster long-term transcriptional memory in primed cells^{21,22}. Consistent with that, we observed that Gal1p was reliably detected in primed cells (Fig S6C). However, both the wild-type and the *rrp6Δ* strain present similar levels for Gal1p suggesting that its cytoplasmic accumulation does not explain the observed differences in transcriptional memory between strains.”*

Major points

1) Authors conclude that changes in H3K4me2/3 do not correlate with the observed transcriptional memory. However, the metagene plots (Fig. 3 C & D) used to draw this conclusion are for all genes. Authors do provide metagene plots for gene subsets (Fig. S3D) and conclude no obvious changes (qualitative conclusion?). However, these data raise two questions. First, the data in Figure S3D suggests that the average level of H3K4 methylation over inducible genes (i.e. that should be unexpressed) is very similar to that over repressible genes (i.e. that should be expressed). This is a surprising observation and raised concerns about how this sub setting was performed. The authors should provide plots for H3K4me2 and H3K4me3 signal for representative genes with established transcriptional status in glucose to confirm that transcribed genes exhibit greater H3K4 methylation than repressed genes. Also, because H3K4me2/3 ChIP-seq signal may vary across these subsets, a comparison of signal for individual genes between conditions is required to evaluate whether significant changes occurred over some, but not all. Are there any genes for which there is a correlation between H3K4 methylation or nucleosome positions and either induction memory or repression memory?

We want to note, that according to our definition “induced genes” do not necessary imply that they have very low expression at time 0, just that their RNA abundance increase (e.g., from “repressed” to “high” expressed but also from “medium” to “high” expressed). Likewise, “repressed genes” also includes those genes that experience only a moderate (but significant) decrease in RNA abundance. As requested by the reviewer, we provide now gene-specific comparison for changes in all chromatin marks (and not only aggregates). We have summarized all that information in the updated Fig. S5 and additional text (marked in red). This allows to identify some subtle differences between groups, but not at the level that we observe for changes in mRNA stability. Please see our reply to reviewer #1.2 for extended discussion.

To convince the reviewer of the quality of our data, we have now plotted our ChIP for representative genes with established H3K4me2/3 status in glucose. We have chosen the representative region displayed in Fig 1 from Weiner *et al.* (PMID 25801168) and added a new panel as figure as Fig S4C showing repressed (SPS22) and highly expressed (PDI1) genes.

For comparison, we extracted the same region from Weiner *et al.* using the SGD browser. Browsable track here:

https://browse.yeastgenome.org/?loc=chrIII%3A41772..50772&tracks=DNA%2CAll%20Annotated%20Sequence%20Features%2Ch3k4me2_0_to_60_mins%2Ch3k4me3_0_to_60_mins&highlight=

2) Authors indicate that changes detected by SLAM-seq reflect changes in cytoplasmic mRNA decay (page 10, lines 21-23). mRNA Turnover is calculated as Nascent Reads/Total Reads, and authors conclude that changes in mRNA Turnover between conditions is an indication of increased cytoplasmic stability or targeted degradation. However, SLAM-seq may reflect both changes in mRNA turnover and increases/decreases in transcription and it is unclear why the authors ignore this component. From the data in the paper, direct comparison of CPM-normalized Nascent reads for each gene in naïve (t_0) vs. primed (t_0') should reveal whether changes in transcription contribute to the observed memory phenomena. The author's ability to infer changes in mRNA stability from SLAM-seq data is dependent on there being no change in transcription. A better way to establish a change in

mRNA turnover rates is to measure them directly, using pulse labeling of 4-tU and measuring the rate of replacement of existing mRNAs with 4tU-labeled RNAs (PMC6152797) .

We fully agree with the reviewer that our current implementation of SLAM-Seq assumes a steady state between transcription rate and mRNA decay. We used that approach, as we reasoned that both t0 and t0' cells would have been growing at least 3h in presence of YPD with no external perturbation and thus would likely fulfil the assumption of steady state for mRNA turnover (transcription=decay). For that reason, in our original submission, we only discussed in detail the measures in YPD conditions. Additionally, when possible, we refer in the manuscript to mRNA turnover (and not to mRNA stability).

Initially we did not explore the changes during early response for two main reasons:

- Upon change to galactose, there is a known massive decrease in transcription (and thus of generation of new transcripts) (Garcia-Martinez et al Mol Cell 2004, PMID: 15260981). This shut down in transcription will massively decrease our ability to measure half-lives (as almost no new 4sU labelled molecules) are generated. This was clear also from our SLAM-Seq measure at t₃₀ and t_{30'}.
- Naïve and primed cells have different velocities for the transcriptional remodelling following galactose addition. We reasoned that those differences, could have secondary effects in our SLAM-Seq measure and thus limit our ability to study the phenomenon. On the contrary, by restricting to YPD conditions in steady-state, we could control better any potential differences. In a condition where the transcriptomic differences are minimal.

To make those points clear to the reader we have modified the section in the manuscript discussing RNA stability (in red).

Independent of that, and to address the reviewer's concern, we have repeated our mRNA stability analysis using a pulse-chase approach. We have focus on the changes upon galactose induction for naïve and primed cells that could not be studied with our previous experimental set up. As our previous experiment shows that the incorporation of 4tU-labeled RNAs during the 30 minutes after changes from glucose to galactose is negligible (e.g., t₃₀ and t_{30'} in Fig S7A), we measured instead the disappearance of 4tU-labeled RNAs (generated during the 60 minutes before the change to YPGal). Specifically, we used a modification of the SLAM-Seq approach with a pulse-chase strategy that does not rely in bead-based RNA enrichment. This strategy enables us to measure the disappearance of the different mRNAs independent of the transcription itself. Reassuringly, this alternative approach to measure mRNA stability during the first 30 min or transition to YPGal suggest gene-specific changes in mRNA turnover similar to our steady-states measures in YPD. We added a new figure (Fig. S7F-J) and modified text to this new version:

To confirm that that mRNA stability differences measured between naïve and primed conditions, was also present during the galactose response we measured mRNA stability using a pulse-chase approach. We labelled RNA for 1h in naïve and primed condition and measured its disappearance upon transition to galactose for 30 minutes (Fig. S7F). This revealed changes in mRNA turnover similar to the group-specific regulation described using a steady-state approach in YPD (Fig. S7G-J).

3) Fig 4 suggests that Mtr4 and Nab3 binding was detected to the vast majority of expressed mRNA in your growth conditions: 791/882 induced, 996/1067 repressed, 3200/? unchanged mRNAs. This is surprising as the canonical binding of these proteins is to a subset of mRNAs, CUTs, snRNAs and snoRNAs. This raises concerns about the background of this method and suggests that additional controls are necessary to establish the biological significance of these interactions. Are there sequence motifs in the RNAs that are recovered compared to an input or tagless control? Are ncRNAs/CUTs enriched in these CRAC experiments?

We want to clarify that what we show in Fig. 4 is the relative association of the different factors to the mRNAs. The fact that a gene is detected only means that there were enough reads for us to consider its analysis (independent of its affinity for the RBP, as mRNA abundance directly affects that measure). Next, we compared the relative binding to Mtr4 and Nab3 normalized by its binding to RNAPol II. For all those analyses, we used the data from Bresson *et al.* from the Tollervey lab who previously developed the CRAC approach. To make that point clearer to the reader, we have modified the main text.

We used previously published CRAC (crosslinking and cDNA analysis) data obtained in naïve conditions measuring the intrinsic association of specific mRNAs to the TRAMP^{30,37} (i.e. Mtr4, Trf4, Trf5) NNS³⁰ (i.e. Nab3) and the SKI complex³⁸ (i.e., Ski2) normalised by their binding to RNA Polymerase II (Supplementary Data S3).

We did not extend our analysis of the quality of CRAC data for components TRAMP and NNS complexes, as it has been previously published independently by 2 different laboratories.

- Bresson, S., Tuck, A., Staneva, D. & Tollervey, D. Nuclear RNA Decay Pathways Aid Rapid Remodeling of Gene Expression in Yeast. *Mol Cell* 65, 787-800.e5 (2017).
- Nues, R. van *et al.* Kinetic CRAC uncovers a role for Nab3 in determining gene expression profiles during stress. *Nat Commun* 8, 12 (2017).

Independent of that, we agree with the reviewer that CRAC (or any other currently available RNA-IP method) is intrinsically noisy (please see our reply to reviewer #1.3). Thus, to further support our working model, we have included in this revised manuscript two orthogonal validations: investigation of protein changes for mRNA decay machinery between naïve and primed conditions and a more detailed investigation of mRNA decay in multiple mutants.

Clarifications and minor concerns

1) Clarify how TMscore is calculated and include it in the main text as well as the figure legend where it is first introduced. Clarity is imperative because the TMscore, which is itself a ratio, is later used again as a ratio of ratios (for instance, see 2 below).

We have clarified how the TM_{score} was measured and indicate it clearly in the main text, methods, and figure legend (in red). The TM_{score} is a relative measure of how strong is gene induction or repression after galactose addition comparing primed vs naïve cells. Specifically, we compared the \log_2 fold changes ($lfc^{15min} - lfc^{3h}$). We have expanded description in the main text, methods and Fig. 2 legend (in red). We have also replotted Fig 2D.

2) The same expression criteria are used to described different sets of genes (page 7, lines 5-8). ($TM_{score} rrp6\Delta / TM_{score} wild-type < 0.667$) is used to described 7 genes with attenuated activation memory as well as 158 genes with enhanced repression memory. While I believe induced and repressed categories were subset based on these criteria, these details should be explicitly stated for the reader.

We apologise for the typo. We went over our original analysis and updated in text used criteria for differential memory between strains. This is now defined as $TM_{score} rrp6\Delta - TM_{score} wild-type < -0.58$.

This criterion is used to define both the attenuated induction memory and enhanced repression memory. For induction memory, the higher the TM_{score} , the stronger the memory. For repression memory, the situation is the opposite. Thus, if the TM_{score} in *rrp6* Δ is much smaller than the wild type, this indicates a stronger repression memory in *rrp6* Δ .

3) In the methods, please provide relevant details for CRAC and subsequent analysis, even if identical to Reference 29. At a minimum, provide: number of cells crosslinked, crosslinking-time, antibodies used, amount of RNA used for library prep, library prep kit used, sequencing platform and read-length. In addition, provide details for how the analysis was performed for data illustrated in Figure 4. For instance: how many reads were required for a given gene to be included in the plot (minimum threshold)?

We apologize for the confusion. As previously discussed, we used published CRAC datasets generated by the Tollervey group. Although we already mentioned that fact in our original manuscript, we made that point clearer for the reader in this revised version. We have now added a methods section detailing how all CRAC data was reanalysed (in red).

4) Include the wild-type strain and a legend in Fig 1C.

Done.

5) For Figure 1A, consider using a different color scheme. The one used in the original manuscript is going to be difficult to see for red-green colorblind readers.

Done.

6) For Figure 2A, two suggestions: first, the colors are not intuitive to represent expression with a dark color and repression with a light color. Second, it would be very helpful to add a plot of the mean expression (as in panel C) for all 6 categories of genes.

We re-plotted the Fig2A following the reviewer's suggestion. We also represent mean RNA expression for each group in new Figure S3F.

For second suggestion, it make a lot of sense to demonstrate the mean expression of each group, we plotted this value with boxplot and put the figures in FigS3D.

7) 540 genes were differentially expressed in wild type vs *rrp6Δ* in naïve conditions. Please categorize these genes with GOterm enrichment and conclude how much, if any, of the memory genes seen are a result of the differential expression of the strains themselves.

We have categorized upregulated and down regulated genes in response to *rrp6* deletion. There are 320 upregulated genes in *rrp6Δ* (compared to wild type). Those genes are enriched in sporulation, developmental process, and membrane assembly. On the other hand, there are 220 downregulated genes in *rrp6Δ*. Those genes are enriched in small molecule catabolic process, cytoplasmic translation and monocarboxylic acid metabolic process. We present those analysis in new Fig S2K-L.

Next, we computed the overlap between differential expressed genes in *rrp6Δ* and the different transcriptional memory groups. The 320 upregulated genes in *rrp6Δ* significantly overlap with the genes classified with induction memory (Hypergeometric test, $p\text{-value} = 4.7 \cdot 10^{-8}$) and with induction memory enhanced by *RRP6* depletion ($2.6 \cdot 10^{-8}$). While the 220 downregulated genes in *rrp6Δ* significantly overlap with the genes classified with repression memory ($p\text{-value} = 0.003$) and repression memory enhanced by *RRP6* depletion ($2 \cdot 10^{-7}$). We indicate that in the revised text (in red, page 6-7).

Independent of this overlap, is important to note that the transcriptional memory phenotype is measured within each strain (and thus controls for the expression at time 0 for the naïve cells). Thus, the differences in naive cells are likely synergistic with the effects on transcriptional memory caused by the depletion of *RRP6*.

8) When calculating TMscore, explain why an absolute value of 5 was added or subtracted from induced and repressed memory genes, respectively (page 21, lines 6 &7)? The plotted values of TMscore (Fig 2D) do not suggest ± 5 .

Initially we added we artificially added 5 to induced genes and artificially subtract 5 for repressed genes to make the plots more intuitive (showing positive values for induction and negative for repression). However, we realise that that was not as intuitive as we initially thought. Thus, we eliminated the ± 5 correction, replot Fig 2D and expanded the text describing the TM_{score} .

Reviewer #3 (Remarks to the Author):

In this very interesting manuscript, the authors show that in addition to transcriptional memory happening at chromatin level that have been previously described, post-transcriptional memory contributes to the adaptation of *S cerevisiae* to a changing environment. They developed a genome wide screening system based on a reporter using GFP under the control of GAL1 promoter and transformed it in the deletion library : it appears as an ingenious and robust tool to highlight factors involved in galactose adaptation memory. After this screening, they pointed out some factors from the TRAMP and nuclear exosome complex, as enhancing memory. Then, they generated data from global methods including RNA seq and SLAM-seq, and used published RNA interactome obtained by CRAC for the factors of interest to understand the contribution of RNA processing to post-transcriptional memory. I believe the results presented here bring new conclusions about adaptation memory, will be of general interest and should be published. However a few points would need to be discussed/addressed before publication :

In the last part of their results, the authors suggest that changes in cytoplasmic mRNA stability contribute to the transcriptional memory phenotypes. Even if the SLAM seq data supports this

hypothesis, I am not sure there is enough evidence to affirm this. They could probably use available data to strengthen their argument:

We thank the reviewer for his/her positive comments and suggestions.

-Could they discuss how cytoplasmic rRNA processing factors were behaving in the reporter assay? We would expect to find Xrn1 and the Ski complex in the memory enhancing factors. Do they find the other subunits of TRAMP and exosome?

Unfortunately, our screen did not provide sufficient coverage to investigate XRN1 (the strain was not present at a detectable level in our transformed pool). Regarding the Ski complex, for example SKI8 was identified as a candidate gene enhancing memory (Fig S1D), while SKI2 was also enhancing memory, but did not reach our threshold for inclusion ($p\text{-adj}=0.12$). Neither NAB3 or MTR4 were at detectable level in our screen. And other components of the TRAMP or NNS complex were not significant hits in either direction. All data was provided in Supplementary Dataset S1. However, to make that information more accessible to the reader, we highlight the case of XRN1 in the revised discussion and point where the data can be explored:

However, we did not capture XRN1 in our initial screen as that strain was below our detection limit (Supplementary Dataset S1).

-CRAC data in naive conditions are available for the SKI complex (Tuck et al, 2013). Could they do a similar analysis than they did with Mtr4? Same for other processing factors as Trf4 and Trf5 (CRAC is available too : Delan-Forino et al, 2020). It would be interesting to check the dependance of these transcripts on Pap2 and Trf5 binding. Pap2 and Trf5 behaviour being different on mRNAs, it would be a good control to assess specificity of Pap2.

We have expanded our CRAC analysis including the datasets that the reviewer suggested. We have plotted the (Trf4/Pap2) and Trf5 as well as the Ski complex in Fig 4. To make our analysis more robust we have increased our threshold for analysing gene (> 20 reads). This increased stringency of our lead to a change for the case of Mtr4 from no significant in the previous version to significant (Fig 4P). Our new analysis revealed that that Pap2 (Trf4) and Trf5 behaviours was similar. However, we noted a difference in respect to Mtr4 for genes with repression memory enhanced by RRP6 depletion (Fig 4P and text in red).

-The published work (Delan-Forino et al 2017) discriminate mRNA subjected to nuclear degradation and the ones subjected to cytoplasmic degradation. Could the authors look at how the top mRNAs sensitive to RNA processing behave regarding post-transcriptional memory to assess contribution of both cytoplasmic and nuclear memory.

To characterizer nuclear vs cytoplasmic degradation, we used the data provided by Delan-Forino *et al* 2017 PLoS Genetics (Supplementary Table S5). There the authors provide information for 200 higher expressed genes regarding for Rrp44 and Rrp44(Rrp41-channel). We define 13 genes (Rrp44_Rrp41wt/-channel value < 1) as prone to nuclear decay and 92 genes (value > 3) as prone to cytoplasmic decay (as defined in the cited publication). The nuclear decayed genes are significantly enriched genes with enhanced repression memory ($p\text{-value} = 2 \cdot 10^{-5}$) while genes preferentially using cytoplasmic decay genes are enriched both in *rrp6Δ* enhanced repression memory ($p\text{-value} = 0.002$) and repression memory not changed by RRP6 depletion ($p\text{ value} = 6 \cdot 10^{-10}$). Please see tables below.

	A	B	C	D	E	F	G	H
1	Table S5: Rrp44 and Rrp44(Rrp41-channel) RPKM for mRNAs							
2		Rrp44_1 (RPKM)	Rrp44_2 (RPKM)	Rrp44_Rrp41-channel_1 (RPKM)	Rrp44_Rrp41-channel_2 (RPKM)	Rrp44 (average RPKM)	Rrp44_Rrp41-channel (average RPKM)	Rrp44_Rrp41wt/-channel
3	DBP2	115.774	144.636	196.748	319.078	130.205	257.913	0.50484078
4	BDF2	95.246	360.169	383.823	342.497	227.7075	363.16	0.627017017
5	RPL9B	503.733	696.832	895.12	953.634	600.2825	924.377	0.649391428
6	RPS14B	649.929	879.62	1034.27	1145.697	764.7745	1089.9835	0.701638603
7	RPS9A	334.101	689.58	770.098	661.1	511.8405	715.599	0.71526162
8	HPT1	301.069	349.082	535.691	336.645	325.0755	436.168	0.74529883
9	RPL7B	296.484	443.569	514.061	469.014	370.0265	491.5375	0.752794039
10	RPL18B	190.607	231.945	318.3	189.918	211.276	254.109	0.831438477
11	TYE7	148.176	218.662	146.804	275.64	183.419	211.222	0.868370719
12	MCR1	172.191	221.182	222.671	196.313	196.6865	209.492	0.938873561
13	BAP2	127.535	211.612	143.592	217.596	169.5735	180.594	0.938976378
14	NRD1	183.095	279.335	234.307	238.254	231.215	236.2805	0.978561498
15	HRP1	106.645	159.346	143.592	126.623	132.9955	135.1075	0.984368003
16	NPL3	399.22	532.573	440.945	438.513	465.8965	439.729	1.059508243
17	LEU4	151.432	99.713	58.213	167.236	125.5725	112.7245	1.113976997
18	RPL18A	243.018	260.987	249.822	196.006	252.0025	222.914	1.130492028
19	ADE12	172.138	98.896	112.672	123.117	135.517	117.8945	1.149476863

However, we suspect that the observed results are likely driven by the fact that those are high expressed genes (and not necessarily due to their nuclear/cytoplasmic decay). For that reason, we decided to keep this information here, but not add it to the main text.

Nuclear decay		total	expected	observed	pvalue	enrich	over
Induction Memory	Attenuated	7	0,0166	0	0,98	under	0
Induction Memory	no changed	451	1,07	0	0,326	under	0
Induction Memory	enhanced	88	0,208	1	0,19	over	4,79
	Induced no memory	336	0,79	0	0,438	under	0
	no change	3526	8,37	5	0,05	under	1,67
	repressed no memory	294	0,698	0	0,48	under	0
repression memory	Attenuated	4	0,009	0	0,99	under	0
repression memory	not changed	611	1,45	2	0,43	over	1,38
repression memory	enhanced	158	0,37	5	0,00002	over	13,33

Cytoplasmic decay		total	expected	observed	pvalue	enrich	over
Induction Memory	Attenuated	7	0,11	0	0,88	under	0
Induction Memory	no changed	451	7,58	4	0,11	under	1,89
Induction Memory	enhanced	88	1,48	0	0,22	under	0
	Induced no memory	336	5,6	1	0,02	under	5,65
	no change	3526	59	48	0,01	under	1,23
	repressed no memory	294	4,9	2	0,12	under	2,47
repression memory	Attenuated	4	0,06	0	0,93	under	0
repression memory	not changed	611	10	20	0,002	over	1,95
repression memory	enhanced	158	2,6	17	6E-10	over	6,4

Clarification needed:

-Fig 1C is unclear to me. Are red and black lines 2 independent replicates? Is Y axis representing the ratio with WT expression? If yes it should be stated.

The red line and black line represent gene expression pattern of two different strains. We have now revised the figure and clarified the legend.

-Is FigS2F about genes with induction memory in WT or delta Rrp6 because it is called for both in the text.

FigS2F refers to genes with induction memory in WT. We have changed the figure legend to make that point clear to the reader.

-Fig S2D,E,F,G : Are all the GO terms found for each category of genes (repressed, induced, ...) represented in the figure or just the Top 16 or 17 GO terms are shown? not clear.

Only the top 17 terms are shown due because of lack of space. The complete list of enriched GO terms are listed in supplementary table S2. We revised figure legend to make that point clear to the reader.

Minor changes:

-Fig 2C is not cited in the text

-p3 l25 little m in mCherry

-p14 l13 Majuscule I in "In" to remove

-when several papers are cited together, it happens several times in the manuscripts that the numbers are not separated by comma.

We have fixed those problems.

REVIEWERS' COMMENTS

Reviewer #1 (Remarks to the Author):

The authors made important efforts to satisfy the requests of the reviewers. The proteomic analyses are a great improvement towards the understanding of the underlying mechanism. These additional data of course also raise novel questions that could be addressed in future studies. In my opinion, the revised manuscript deserves to be published in Nature Communications after some minor changes.

1) Figures 4 and 5 would benefit from a reduction in the number of panels. The Mass Spectrometry data, which are probably key for understanding the mechanism, are contained in panels U and V (almost the whole alphabet!) of Figure 4. The NNS complex, which is less tackled along the work, could be moved to Supplementary material. Same for Figure 5, in which ski2 Δ could be moved to Suppl. The authors are free to choose what to move to Suppl., but this may be one way to help the reader focus on the most important informations. It should nevertheless be mentioned that the Reviewers asked for quite a lot of detailed analyses that the authors overall satisfied. However, it should be possible to present these data in a more synthetic way to produce a more balanced manuscript.

2) Since the revised manuscript is now proposing a molecular mechanism, the drawing of a model would greatly help to summarise the main message of the publication.

Minor comment: there seems to be a mistake in the labelling of Figures 3A and 3B. Shouldn't it say N P N P instead of N N P P?

Reviewer #2 (Remarks to the Author):

The authors have, for the most part, addressed my concerns and responded to my suggestions. The manuscript is now easier to understand and more convincing.

Reviewer #3 (Remarks to the Author):

I think the manuscript has been significantly improved since first version and most my comments has been addressed and I consider it as ready to be published. I expect these high quality results will be of considerable use for the community.

REVIEWERS' COMMENTS

Reviewer #1 (Remarks to the Author):

The authors made important efforts to satisfy the requests of the reviewers. The proteomic analyses are a great improvement towards the understanding of the underlying mechanism. These additional data of course also raise novel questions that could be addressed in future studies. In my opinion, the revised manuscript deserves to be published in Nature Communications after some minor changes.

The thank the reviewer for his/her positive comments.

1) Figures 4 and 5 would benefit from a reduction in the number of panels. The Mass Spectrometry data, which are probably key for understanding the mechanism, are contained in panels U and V (almost the whole alphabet!) of Figure 4. The NNS complex, which is less tackled along the work, could be moved to Supplementary material. Same for Figure 5, in which ski2 Δ could be moved to Suppl. The authors are free to choose what to move to Suppl., but this may be one way to help the reader focus on the most important informations. It should nevertheless be mentioned that the Reviewers asked for quite a lot of detailed analyses that the authors overall satisfied. However, it should be possible to present these data in a more synthetic way to produce a more balanced manuscript.

We moved the suggested panels to supplementary material as suggested by the reviewers.

2) Since the revised manuscript is now proposing a molecular mechanism, the drawing of a model would greatly help to summarise the main message of the publication.

We added a model of the work in the new Figure 6.

Minor comment: there seems to be a mistake in the labelling of Figures 3A and 3B. Shouldn't it say N P N P instead of N N P P?

We have corrected the typo in the figure.

Reviewer #2 (Remarks to the Author):

The authors have, for the most part, addressed my concerns and responded to my suggestions. The manuscript is now easier to understand and more convincing.

The thank the reviewer for his/her positive comments.

Reviewer #3 (Remarks to the Author):

I think the manuscript has been significantly improved since first version and most my comments has been addressed and I consider it as ready to be published. I expect these high quality results will be of considerable use for the community.

The thank the reviewer for his/her positive comments.